# Reconstructed natural runoff helps to quantify the relationship between upstream water use and downstream water scarcity in China's river basins

Xinyao Zhou[1], Yonghui Yang[1], Zhuping Sheng[2], Yongqiang Zhang[3]

[1]Key Laboratory of Agricultural Water Resources, Hebei Laboratory of Agricultural Water-Saving, Center for Agricultural Resources Research, Institute of Genetics and Developmental Biology, Chinese Academy of Sciences, Shijiazhuang 050021, China
[2]Texas A&M Agrilife Research Center, El Paso, Texas 79927, USA
[3]Key Laboratory of Water Cycle and Related Land Surface Processes, Institute of Geographic Sciences and Natural Resources Research, Chinese Academy of Sciences, Beijing 100101, China

*Correspondence to*: Yonghui Yang (yonghui.yang@sjziam.ac.cn)

**Abstract.** The increasing conflicts for water resources between upstream and downstream regions appeal for chronological insight across the world. While the negative consequence of downstream water scarcity has been widely analysed, the quantification of influence of upstream water use on downstream water scarcity has received little attention. Here non-anthropologically intervened runoff (natural runoff) was first reconstructed in upstream, middle stream and downstream regions in China's 12 large basins in the 1970s to 2000s time period using the Fu-Budyko framework, and then compared with the observed data to obtain the developmental trajectories of water scarcity, including water use to availability (WTA) and per capita water availability (FI) in decadal scale. Furthermore, a contribution analysis was used to investigate the main drivers of water scarcity trajectories in those basins. The results show that China as a whole has experienced a rapid increase of WTA stress with surface water use rapidly increasing from 161 billion $m^3$ (12% of natural runoff) in 1970s to 256 billion $m^3$ (18%) in 2000s, approximately 65% increase occurring in North China. In 2000s, the increase of upstream WTA stress and the decrease of downstream WTA stress occurred simultaneously for semi-arid and arid basins, which was caused by the increasing upstream water use and the consequent decreasing surface water use in downstream regions. The influence of upstream surface water use on downstream water scarcity was less than 10% in both WTA and FI for humid and semi-humid basins during the study period, but with an average of 26% in WTA and 32% in FI for semi-arid and arid basins, and the ratio kept increase from 10% in 1970s to 37% in 2000s for WTA and from 22% in 1980s to 37% in 2000s for FI. The contribution analysis shows that the WTA contribution greatly increases in 2000s mainly in humid and semi-humid basins while decreases mainly in semi-arid and arid basins. The trajectories of China's water scarcity are closely related to socioeconomic development and water policy changes, which provides valuable lessons and experiences for global water resources management.

**Keywords:** Water scarcity, Natural runoff reconstruction, Upstream-downstream relationship, Quantitative analysis, China

# 1 Introduction

Water scarcity is one of major challenges for hampering the United Nations sustainable development goals. This is particularly important for downstream areas where upstream water inflow is needed to satisfy downstream water demand exceeding local-generated water resources. It was estimated that up to 1 billion people would have water scarcity problem if upstream water was not provided for downstream areas (Oki et al., 2001). Upstream drought and excessive water use would exacerbate downstream water scarcity, causing the consequent cooperative or conflictive events (Munia et al., 2016). These facts make it critical to understand the influence of upstream water use on downstream water scarcity under a changing climate and with intensified human activities.

Many studies have been conducted to analyse the negative impacts of upstream water use on downstream environment (Poff et al., 2007; Arfanuzaman and Syed, 2018), biology (Brown and King, 2006; Petes et al., 2012), water quality (Dodds and Oakes, 2008), and socioeconomic issues (Jack, 2009; Nordblom et al., 2012; Al-Faraj and Tigkas, 2016). Despite the widespread recognition of the negative impacts, only limited quantitative research studies have been performed to unravel the upstream-downstream interactions on water resources and water scarcity. Munia et al. (2016) simulated water use and water availability using the PCR-GLOBWB (PCRaster Global Water Balance) model in global transboundary river basins in 2010, and found that 288 out of 298 middle-stream and downstream sub-basin areas experienced some change in stress level after accounting for upstream water use, affecting 0.29-1.13 billion people in transboundary river basins. Veldkamp et al. (2017) used global multi-model assessment to examine the impact of different human interventions (HI) on monthly water scarcity over the period 1971-2010. Their results showed that HI was the main driver of water scarcity, aggravating water scarcity for 8.8% of the global population but alleviating it for another 8.3%. Positive impacts of HI mostly occur upstream, whereas HI aggravates water scarcity downstream. Duan et al. (2018) investigated the water availability and water stress over the conterminous United States (CONUS) from 1981 to 2010 using statistical water use data and simulated water supply using the WaSSI (Water Supply Stress Index) model. They found that 12% of the CONUS land relied on upstream incoming flow for adequate water supply, while local water alone was sufficient to meet the demand in another 74% of the area. Munia et al. (2018) developed a framework to quantify the dependency of downstream water stress on upstream water supply and applied the framework to global transboundary river basins. Surprisingly, they found that the majority (1.15 billion) of those people (1.18 billion) currently suffer from water stress only because they excessively use water within each basin and the water use from upstream does not have significant impact on the downstream stress status. These studies, however, either focusing on transboundary river basins or dependency analysis, indirectly indicated the importance of upstream water inflow to downstream water scarcity. There is a great need for direct quantification of influence of upstream water use on downstream water scarcity in river basins as a whole.

As one of three countries with greatest water risk hotspots, China is facing serious water stress, especially in its northeastern regions (OECD, 2017). Meanwhile, the downstream environment has been severely deteriorated in some arid basins (Li et al., 2013; Lu et al., 2015; Zhao et al., 2016). Therefore, this study selected China to quantify the impact of upstream water

use on downstream water scarcity. Understanding the past trajectories of China's water scarcity in upstream and downstream catchments and quantifying the relationships between upstream water use and downstream water scarcity can help to better define pathways to future sustainability, avoid further irreversible environmental degradation, and address future challenges of climate change and human interventions.

Water scarcity can be divided into two aspects: water availability and water use (Kummu et al., 2016). Water availability induced scarcity is a "demographic-driven scarcity" when a large population compete for limited water resources, leading to disputes. Water use caused scarcity refers to a "demand-driven scarcity" when excessive water is consumed due to socioeconomic development but irrelevant to the population (Falkenmark, 1997; Kummu et al., 2010). The combined use of the two aspects can therefore provide a complete picture to describe water scarcity.

It is difficult to get long-term water use and the related water scarcity data in China due to data inaccessibility. As a substitution, the gap between observed runoff and modelled non-anthropologically intervened runoff (hereafter called natural runoff) can be treated as surface water use. There are numerous studies on natural runoff driven by process-based models such as VIC (Variable Infiltration capacity) (Wang et al., 2010; Chang et al., 2015), WBM (Water Balance Model) (Guo et al., 2017), ORCHIDEE (Organizing Carbon and Hydrology in Dynamics Ecosystems) (Piao et al., 2007), and SWAT (Soil and Water

Assessment Tool) (Luo et al., 2016). However, difficulties in calibrating complex parameters limit model application to one or a few basins (Zhang et al., 2007; Jiang et al., 2015; Zhai and Tao, 2017). In comparison, Budyko framework is widely used at an annual to decadal scale and in a large spatial scale (Zhang et al., 2001; Zhang et al., 2009;Zheng et al., 2009). Six Budyko framework models were tested here and eventually the one-parameter Fu-Budyko model was used to reconstruct natural runoff in the catchments because of its optimal performance (Fu, 1981). Fu-Budyko model has also been successfully validated

across the globe (Teng et al., 2012; Zhou et al., 2012; Li et al., 2013; Du et al., 2016). As such, this study used this model to reconstruct decadal natural runoff for the period of 1961–2010 in upstream and downstream regions within 12 large basins in China, which cover over 50% of mainland China.

In this study, we aim to answer following three questions, and provide experiences and lessons for global water resources management. They are:

i.    How surface water scarcity developed in upstream and downstream regions of the selected basins in China during the past decades;

   ii.    How to quantify the influence of upstream water use on downstream water scarcity; and

  iii.    What is the main factor that has driven China's water scarcity change.

## 2 Materials and Methods

### 2.1 Materials

#### 2.1.1 Hydrological data

Since digital runoff data is limited in China, we obtained runoff data from the following two-sources: official sources in Hai and Shiyang River Basins and published literature (Table 1). The reliability of the published annual runoff data was verified based on the following two criteria. First, for a specific gauge station, at least two related published data sources of overlapping study periods were prepared. Then the annual runoff data was extracted and a cross validation conducted to limit errors below 5%. Second, the published annual runoff data were further verified by comparing the trends in the processed data and in other data published coincidentally, such as published work for Dongting lake by Yang et al. (2015), for Huangpu river by Shi and Wang (2015) and so on.

**Insert Table 1 here**

The annual runoff measured in a total of 132 gauge stations was verified. Based on the record length and spatial distribution of the data, 37 gauge stations that are representative for upper, middle and lower reaches were used in this analysis. While the length of data from 29 out of 37 basins spanned for an entire period of 1961–2010, data from other 8 basins spanned for over 40 years. The basin boundaries were based on the delineations in "Data Sharing Infrastructure of Earth System Science" (http://www.geodata.cn/) and sub-basin boundaries were delineated in ArcHydro tool (Fig. 1).

**Insert Figure 1 here**

#### 2.1.2 Climatic factors

Gridded monthly precipitation and temperature (maximum, minimum and mean temperature) for 1961–2010 were downloaded from "China Meteorological Data Sharing Service System" (http://cdc.nmic.cn/). The spatial resolution of the gridded dataset is $0.5° \times 0.5°$. The gridded datasets were produced by merging resampled GTOPO30 dataset and interpolated climate data from 2472 stations using the TPS (Thin Plate Spline) method. Also daily climate data at point-scale (maximum and minimum temperature, wind speed, relative humidity and sunshine hours) from 563 national weather stations for the period 1961–2010 were downloaded from the same website.

#### 2.1.3 Population data

The population data from Gridded Population of the World (GPW) (http://sedac.ciesin.columbia.edu/data/collection/gpw-v4) was used to estimate the basin-scale population. Given the limitation of the data record length, the GPW data for 1990, 2000 and 2010 were respectively used to get the population for the 1980s, 1990s and 2000s. The resolution was ~5 km for 1990 and 2000 datasets and ~1 km for 2010 dataset.

## 2.2 Methods

### 2.2.1 Fu-Budyko framework

The Fu-Budyko framework is expressed as:

$$F(\varphi) = 1 + \varphi - \left(1 + \varphi^{\theta}\right)^{1/\theta} \tag{1}$$

where $F(\varphi)$ is evaporation ratio, $\varphi$ is the Aridity index (AI), calculated from ratio of potential evapotranspiration ($ET_0$) to precipitation ($P$) on annual scale, the $\theta$ parameter is related to catchment characteristics with the range of $1\sim\infty$. In this study, AI of each catchment was calculated at mean annual scale for the period of 1961-2010 and the catchments were classified into humid, semi-humid, semi-arid and arid for AI ranging from 0.375~0.75, 0.75~2, 2~5 and 5~12, respectively (Ponce et al., 2000; Arora, 2002). Annual natural runoff was calculated in unit of mm/year as $P*(1-F(\varphi))$, and then changed into discharge in units of $10^9$ m$^3$/year by multiplying the catchment area.

Studies have shown that anthropologic interventions had intensified across China since the 1980s, driven by the economic reform and the open door policy (Yang and Tian, 2009; He et al., 2013; Jiang et al., 2015). We therefore assumed that the observed runoff for 1961–1970 was natural and not (or less) disturbed by human activities. Using the observed $ET_0$, $P$ and observed discharge, the parameter $\theta$ was calculated using the least-square data fitting method for the period 1961–1970, then the fitted parameter was used to reconstruct decadal natural runoff for the period 1971–2010.

### 2.2.2 Estimation of $ET_0$ and $P$

Two equations – Hargreaves (HG) and Penman-Monteith (PM) – were used to estimate $ET_0$ (Allen et al., 1998). The HG-$ET_0$ was based on a gridded dataset at monthly scale while PM-$ET_0$ was based on a pointed dataset at daily scale. The PM equation ranked as the best equation for estimating $ET_0$ but the sparse distribution of climate stations limited its application in western China. The continuous spatial coverage of the gridded dataset can provide full estimation of HG-$ET_0$ in western China. However, large discrepancies between HG-$ET_0$ and PM-$ET_0$ were found in different regions over the world in previous studies (Temesgen et al., 2005; Gavilan et al., 2006; Trajkovic, 2007; Bautista et al., 2009; Sivaprakasam et al., 2011; Berti et al., 2014). Thus a more accurate $ET_0$ can be obtained by combining the two estimations.

Hargreaves equation is described as (Allen et al., 1998):

$$ET_0 = 0.0023(T_{mean} + 17.8)(T_{max} - T_{min})^{0.5} R_a \tag{2}$$

Where $T_{mean}$ is the $i$th-month mean temperature; $T_{max}$ is the $i$th-month mean maximal temperature; $T_{min}$ is the $i$th-month mean minimal temperature; and $R_a$ is the extraterrestrial radiation for the middle day of the $i$th-month. The standard values of empirical parameters are 0.0023, 17.8 and 0.5. The unit for both $ET_0$ and $R_a$ is mm/day and then $ET_0$ was multiplied by the number of days in the $i$th-month to get monthly $ET_0$. The extraterrestrial radiation $R_a$ is calculated with FAO56 method (Allen et al., 1998).

FAO56 Penman-Monteith equation is described as below (Allen et al., 1998):

$$ET_0 = \frac{0.408\Delta(R_n - G) + \gamma \dfrac{900}{T + 273} u_2 (e_s - e_a)}{\Delta + \gamma(1 + 0.34 u_2)} \tag{3}$$

where $R_n$ is the net radiation at the crop surface [MJ m$^{-2}$ day$^{-1}$], $G$ is the soil heat flux density [MJ m$^{-2}$ day$^{-1}$], $T$ is the mean daily air temperature at 2 m height [°C], $u_2$ is the wind speed at 2 m height [m s$^{-1}$], $e_s$ is the saturated vapour pressure [kPa], $e_a$ is the actual vapour pressure [kPa], $e_s$-$e_a$ is the vapour pressure deficit [kPa], $\Delta$ is the slope of vapour pressure-temperature curve [kPa °C$^{-1}$], $\gamma$ is the psychrometric constant [kPa °C$^{-1}$]

The monthly gridded HG-$ET_0$ and daily pointed PM-$ET_0$ were scaled up to an annual value. At the annual scale, HG-$ET_0$ was adjusted by multiplying the gridded coefficient (interpolated by the IDW method) as the ratio of the PM-$ET_0$ to HG-$ET_0$. The gridded annual precipitation was aggregated from the gridded monthly precipitation data and then adjusted by the point-scale data as mentioned above. The basin-scale annual $P$ and $ET_0$ were obtained by weighting the average of grid data within each basin.

### 2.2.3 Water stress and availability

Two indicators – WTA (Water use To Availability) and FI (Falkenmark Index, referring to per capita water availability) – were used for the developmental analysis in surface water scarcity. WTA indicates moderate or high water use stress when over 0.2 or 0.4, respectively (Vörösmarty et al., 2000). FI indicates moderate, high and extreme water availability stresses when it drops below 1,700, 1,000 and 500 m$^3$ cap$^{-1}$ year$^{-1}$, respectively (Falkenmark, 1997). The calculation of WTA was conducted in decadal scale for 1970s, 1980s, 1990s, and 2000s, respectively, while FI was calculated in decadal scale for 1980s, 1990s, and 2000s due to the limited access of population data.

$$WTA = WU / WA \tag{4}$$

$$FI = WA / Population \tag{5}$$

$$WU = \begin{cases} Q_{nat} - Q_{obs} & upstream \\ WU_{local} - WU_{former} & middle / downstream \end{cases} \tag{6}$$

$$WA = \begin{cases} Q_{nat} & upstream \\ Q_{nat} + Q_{in} & middle / downstream \end{cases} \tag{7}$$

$WU$ and $WA$ indicates surface water use and water availability in each decade from the 1970s to 2000s, $Q_{nat}$ and $Q_{obs}$ are natural and observed discharge in the same decade, $Q_{in}$ is the incoming observed discharge from upper reach, $WU_{local}$ and $WU_{former}$ are the surface water use in middle/downstream regions and its former regions, respectively.

For Hai, Shiyang, Hei and Tarim River Basins, natural discharge at the middle and lower reaches was assumed to be the same as the upper reaches or the aggregate discharge from upstream tributaries. This is because most of the water was subsequently consumed and therefore little runoff was generated in the downstream regions (Zhang et al., 2015; Zhang et al., 2016).

It is noted that only nine large basins were selected to analyze past changes in surface water scarcity in all three reaches (upper, middle and lower) because runoff data were not available in the downstream regions of Liao, Huai and Qiantang River Basins. For example, hydrological data at outlet station in Liao River Basin is available in 1984-2010; there were no hydrological data at outlet station in Huai River Basin; streamflow data were only available in tributary stations in Qiantang River Basin. For the above-mentioned three basins, we only used the available data from upper stream or tributaries for estimating WTA and FI.

### 2.2.4 Quantitative analysis

To quantify the influence of upstream water use on downstream water scarcity, an experiment was designed by involving two scenarios: one with upstream water use (S1) and another without upstream water use (S2). In the first scenario (S1), the downstream water availability was the sum of local natural discharge and incoming observed flow; in the second scenario (S2), the downstream water availability was the aggregation of local natural discharge and natural discharge from the upper reaches.

### 2.2.5 WTA Contribution in water scarcity change

The contribution rate of WTA change in water scarcity change is estimated as follow:

$$Contribution_{WTA} = \frac{\Delta WTA}{\Delta WTA + \Delta FI} \tag{8}$$

$$\Delta WTA = \left| zscore(WTA)_i - zscore(WTA)_j \right| \tag{9}$$

$$\Delta FI = \left| zscore(FI)_i - zscore(FI)_j \right| \tag{10}$$

Where $\Delta WTA$ and $\Delta FI$ indicate the absolute difference between two periods i and j in WTA and FI standardized (zscore) by subtracting the mean then dividing by the standard deviation. Every decade is compared with its previous decade to get the stress change, for example, 2000s.vs.1990s, 1990s.vs.1980s and 1980s is set to 0.

## 3 Results

### 3.1 Reliability of Fu-Budyko framework

The reliability of the Fu-Budyko framework in reconstructing annual natural discharge is summarized in Figure 2. The model captures well the fluctuations of observed discharge in both time and space during the calibration period of 1961-1970 in all catchments, with biases (($\sum Q_{sim}$-$\sum Q_{obs}$)/ $\sum Q_{sim}\times100\%$) of 4.8%, 1.2%, 10%, -0.2%, -1.3%, 0.3%, 2.5%, -0.5%, 0.8%, 0.9%, -2.2% and 8.2% for Yangtze, Xi, Min, Qiantang, Huai, Songhua, Yellow, Liao, Hai, Hei, Shiyang and Tarim River Basins respectively (The calculation of biases for most stations using the downstream observed runoff while using upstream

observed runoff for Hai, Hei, Shiyang and Tarim). Increasing gaps between the observed and natural discharge, however, are observed in semi-arid and arid basins, especially the Hai, Hei, Shiyang and Tarim River Basins. These gaps are regarded as water use from anthropologic activities.

**Insert Figure 2 here**

The magnitude of gaps between observed and natural discharge varies in different reaches and different periods as shown in Figure 3. For humid regions with large discharge, the natural discharge is quite consistent with the observed one, leading to small gaps during all study periods in both upstream and downstream regions. However, situations are different for arid basins with small discharge, where the gap between observed and natural discharge in upstream and middle stream regions is relatively small from the beginning of the study period, and increases as time goes by. While in downstream regions, the gap

is large from the beginning, and rapidly increases with time going by, especially in 1980s and 1990s.

**Insert Figure 3 here**

## 3.2 Water scarcity trajectories

### 3.2.1 National range overview

Generally, the surface water has become scarcer from 1970s to 2000s in China, with national WTA increasing from 0.12 to

0.18 and surface water use increasing from 161 billion $m^3$ in 1970s to 256 billion $m^3$ in 2000s (Fig. 4). The 65% increase of surface water use occurs in northern basins, including Songhua, Huai, Yellow, Liao, Hai, Hei, Shiyang and Tarim River Basins. For humid (Xi, Min and Qiantang River Basins), semi-humid (Yangtze, Songhua and Huai River Basins), semi-arid (Yellow, Liao and Hai River Basins) and arid basins (Hei, Shiyang and Tarim River Basins), WTAs have increased from 0.1, 0.1, 0.36 and 0.81 in 1970s to 0.14, 0.15, 0.7 and 0.95 in 2000s, respectively. Meanwhile national FI decreases from 1,534 to 1,265 $m^3$.

Regarding to different climate zones, FI has decreased from 1,943 in 1980s to 1,680 in 2000s for humid basins, and from 239 $m^3$ to 226 $m^3$ for semi-arid basins, but it has increased from 1,740 in 1980s to 1,772 $m^3$ in 2000s for semi-humid basins and from 866 $m^3$ to 1,255 $m^3$ for arid basins.

According to FI, Xi River Basin changed from no stress (1,782 $m^3$ $cap^{-1}$ $year^{-1}$) to moderate stress (1,583 $m^3$ $cap^{-1}$ $year^{-1}$) and Tarim River Basin changed from high stress (942 $m^3$ $cap^{-1}$ $year^{-1}$) to moderate stress (1,467 $m^3$ $cap^{-1}$ $year^{-1}$) from 1980s to

2000s, while the stress level remained almost unchanged for all the other basins. According to WTA alone, stress level changed from low stress (0.14) to high stress (0.55) for Songhua River Basin, from low stress (0.19) to moderate stress (0.27) for Huai River Basin, from moderate stress to high stress for Yellow (0.33-0.7), Liao (0.36-0.43) and Shiyang (0.33-0.71) River Basins from 1970s to 2000s, while the rest remained at their original stress levels.

For most basins, 1980s is a turning point with rocketing WTA. The magnitude of WTA increase is 40% for Yangtze River

Basin, 56% for Xi River Basin, 64% for Songhua River Basin, 52% for Yellow River Basin, 31% for Hai River Basin, 67% for Shiyang River Basin and 50% in national ranges. Meanwhile, FI changed little in the same period. The changes have probably linked to the water use increase because of China's economic reform and the open door policy at the end of 1970s.

**Insert Figure 4 here**

### 3.2.2 Upstream and downstream relationship

Meanwhile, different basins experience different developmental trajectories in water scarcity for upstream, middle stream and downstream regions (Fig. 5). From the WTA perspective, most humid and semi-humid basins, for example the Min, Pearl and Yangtze River Basins, show the non-stress status and the fluctuations in WTA for both upstream and downstream regions during the study periods. Songhua River Basin shows continuous increase in WTA for both upstream and downstream regions, which has led the upstream regions into stressed status in 2000s. For semi-arid and arid basins, the elevated water use has led the upstream regions from no/low stressed status to high stress and the increasing WTA in downstream regions. It is noteworthy that WTA begins to decrease in downstream regions in 2000s (Hei River Basin shows the decrease in middle stream) while its upstream counterpart still increases. The decrease in WTA for downstream regions is caused by the reduced incoming discharge from upstream regions, which forces the downstream water users to exploit groundwater as a supplement source for water supply (Water Resources Bulletin of Hai River Basin, 2015).

**Insert Figure 5 here**

From the FI respective, the decreasing trend is dominant in both upstream and downstream regions. The Yangtze, Min, Songhua, Hei and Tarim River Basins and the upstream of Pearl have no FI stress. FI has largely decreased in downstream regions compared to its upstream counterparts for eastern basins, however, the reverse is observed in western basins. This is driven by the migration during the study period. Since the end of 1990s, the rapid urbanization has formed some metropolis in downstream regions in eastern China, such as Beijing in the downstream of Hai River Basin, Shanghai in the downstream of Yangtze River Basin, Guangzhou in the downstream of Pearl River Basin, leading to population concentration and FI decrease in those regions (Yang and Chen, 2014). However, for northwestern inland basins, big cities are usually located in the middle reach oasis such as Zhangye in middle stream regions of Hei River Basin, or Aksu in middle stream regions of Tarim River Basin. Meanwhile, the exacerbated degradation of the downstream ecological environment has caused downstream inhabitants to migrate to the middle stream. Thus FI generally decreases in middle stream regions while it increases in downstream in northwestern river basins.

### 3.3 Quantifying influence of upstream water use on downstream water scarcity

Scenario analysis shows the quantitative influence of upstream water use on downstream water scarcity (Fig. 6). For humid and semi-humid river basins (Xi, Min, Yangtze and Songhua River Basins), the influence of upstream water use on downstream water scarcity is negligible during the study period, with less than 10% difference in both WTA and FI between two scenarios. The influence of upstream water use on downstream water scarcity rapidly increased in 2000s for Songhua River Basin, with the WTA difference between two scenarios increasing from 12% in 1990s to 27% in 2000s and the FI's impact doubled from around 700 $m^3$ $cap^{-1}$ $year^{-1}$ in 1990s to 1,400 $m^3$ $cap^{-1}$ $year^{-1}$ in 2000s.

**Insert Figure 6 here**

In contrast, upstream water use largely exacerbates downstream water scarcity in semi-arid and arid basins, and the influence of upstream water use on downstream water scarcity continued to increase from 1970s to 2000s. On average, the WTA

impact extent for all the five semi-arid and arid basins increased from 10% in 1970s to 37% in 2000s and the FI impact extent increased from 22% in 1980s to 37% in 2000s. Among the five basins, Tarim River Basin is the largest human-intervened basin with the WTA increasing from 51% in 1970s to 86% in 2000s and FI increasing from 75% in 1980s to 86% in 2000s. Hai River Basin is the fastest scarcity-exacerbated basin with WTA increasing from 7% in 1970s to 87% in 5 2000s and FI increasing from 59% in 1980s to 87% in 2000s.

## 3.4 Driven factors of water scarcity trajectories

The combined analysis of WTA and FI (Fig. 7) shows that the Hai, Yellow, and Shiyang River basins and the middle stream of Hei River Basin are in both WTA and FI stress simultaneously. While the upstream of Pearl River Basin is experiencing FI stress, and the whole Tarim River Basin, the downstream of Hei River Basin and the upstream of Songhua River Basin 10 are in WTA stress driven by agricultural sector (Nian et al., 2014; Feike et al., 2015). The water scarcity trajectories show that the WTA stress is still increasing in downstream of Hei, and Hai river basins and middle stream of Tarim and Yellow River Basins, while decreasing in downstream of Tarim and Yellow River Basins and upstream of Songhua and Hei River Basins. Statistical data in period of 2003-2017 showed that most basins have the decreasing water use in agricultural sector except Songhua (69-84%), Huai (65-67%) and Northwest inland (91-91%) river basins (MWR, 2003-2017). However, 15 agriculture is still the largest water-consuming sector in 2017 with 84%, 67%, 61%, 69%, 67%, 48%, 59%, 45%, and 91% percent in Songhua, Liao, Hai, Yellow, Huai, Yangtze, Pearl, Southeast (Min and Qiantang) and Northwest inland (Shiyang, Hei and Tarim) river basins respectively.

**Insert Figure 7 here**

The contribution analysis shows that as a whole the WTA's influence increases in 2000s in most humid and semi-humid 20 basins (XI, MIN) while decreases in most semi-arid and arid basins (YL, HAI, HEI, SY) (Fig. 8a). For upstream regions in most basins, the WTA's influence becomes less in 2000s (Fig. 8b). On the contrary, in downstream regions, WTA's influence decreases in most humid and semi-humid basins while increases in most semi-arid and arid basins in 2000s (Fig. 8c). This indicates that there are more intensive human interventions in downstream regions, especially those semi-arid and arid basins over China.

25 **Insert Figure 8 here**

## 4 Discussions

### 4.1 Suitability of Fu-Budyko framework

The fitted parameter $\theta$ was greatly influenced by topography. Taking three basins with different climates – humid Yangtze River Basin, semi-arid Yellow River Basin, and arid Hei River Basin – for example, the values of $\theta$ are 1.6, 1.9 and 1.3 30 respectively for upstream regions while those are 1.8, 2.3 and 1.8 respectively for downstream regions. Given the fact that steeper terrains in upstream and flatter terrains in downstream, the values of $\theta$ are probably related to topography. The result

is consistent with that from Sun et al. (2007), who found that three factors – infiltration rate, water storage capacity and average slope – had impact on $\theta$ in Fu-Budyko framework. Other influential factors were also indicated in other studies, such as vegetation cover (Li et al., 2013), aridity index (Du et al., 2016), and soil characteristic (Gerrits et al., 2009).

Note that Fu-Budyko framework was suitable for annual or mean annual studies while the application in finer temporal scale

was restrained. This has been proved by Zhang et al. (2008), who has tested the Budyko model over 265 Australian catchments at different time scales, including mean annual, annual, monthly and daily. They found at the annual scale, the model works well for most of the catchments with 90% of them having values of the coefficient of efficiency greater than 0.5 and less than 3% of the catchments have bias values greater than 10%.

What is more, previous studies proved that Budyko framework performed badly in extremely arid environments where water

systems are typically unclosed with intense human interference and irrigation, e.g., the northwestern China. Here we found that Du et al. (2016) successfully applied a Budyko framework in Hei River Basin by dividing the basin into six sub-basins. They calibrated the model separately in different sub-basins and found the model performed quite well in the upper mountainous regions with little human interventions while the model was almost impossible to validate in downstream sub-basins. Thus, we also divided the arid basins (including Hai, Shiyang, Hei and Tarim River Basins) into upper little

disturbed mountainous sub-basins and downstream intensively interfered sub-basins. The Fu-Budyko framework was directly applied in the mountainous sub-basins.

## 4.2 The link between China's water policies and water use changes

After the end of 1970s when China's economic reform and open-door policy started, economic development was set as the primary goal, leading to rapid economic advancement in the 1980s, with the GDP increasing fourfold from 364.5 billion

RMB in 1978 to 1699.2 billion RMB in 1989 (National Bureau of Statistics of China, 2017). Our study showed that with rapid economic increase, surface water use also rocketed from 79 billion $m^3$ in 1970s to 138 billion $m^3$ in 1980s for the 12 basins, with the increase in surface water use of 25.4 billion $m^3$ (63%) for humid basins, 18.6 billion $m^3$ (120%) for semi-arid basins, 9.8 billion $m^3$ (59%) for semi-humid basins and 5.1 billion $m^3$ (90%) for arid basins. Meanwhile, the increase of surface water use simultaneously occurred in both upstream and downstream regions in this period, and the increase in

magnitude of surface water use was higher in upstream regions from humid to arid basins. In some cases, the expansion of arable land was the main driver for the increase of surface water use (Yang and Tian, 2009). While in another case, due to the lower priority in water allocation the share of agricultural water use decreased from 64% to 35% from 1985 to 2001, leading to industrial sector being the major contributor in water use increase (Lohmar et al., 2003). In summary, the water resources management was fragmented and sector-oriented due to overlapping responsibilities and lack of effective

coordination, leading to rapid increase in surface water use and conflicts between upstream and downstream and different sectors.

Aiming to address conflicts and shortfalls of the deficient and fragmented system that arose during the 1970s and early 1980s, the 1988 Water Law was implemented as the first fundamental legislation regulating water activities (Shen, 2014). By

encouraging utilization of water resources rather than water saving, the law facilitated the booming of thousands of engineering projects but failed to effectively address water shortages and environmental degradation in China's water resources during the period of 1990s (Jiang, 2017). Our study showed that total surface water use continuously increased from 138 billion m$^3$ in 1980s to 178 billion m$^3$ in 1990s, with 23.8 billion m$^3$ increase (36%) from humid basins, 7.4 billion

m$^3$ increase (28%) from semi-humid basins, 6.2 billion m$^3$ increase (18%) from semi-arid basins and 2.6 billion m$^3$ increase (25%) from arid basins. Meanwhile, the surface water use in upstream regions also increased, while downstream regions were divergent with upward trend in humid and semi-humid basins and downward trend in semi-arid and arid basins due to decrease in water availability. Consequently, 1990s was known as a period with the frequent outbreaks of water-related crises, such as the disappearance of the inland Juyanhai Lake of Hei River Basin in 1992 (Jiang, 2017), the annual average of

107 dry days of the main channel of Yellow river in 1990s (CPSP, 2005), the rapid drop of groundwater table in North China Plain (Jia, 2011), and the severe pollution in surface water and ground water in major rivers (Wu et al., 1999). To summarize, the 20-years rapid development and the neglect of environmental issues caused an extremely tense water-human relationship and threatened the human well-beings and regional sustainable development in 1990s.

When entering into 2000s, in view of the failure of the traditional principle of water use, managing water resources in a

sustainable and efficient manner was of increasing significance (Shen, 2014; Jiang, 2017). Reflecting a significant transition in water governance from construction and utilization (project oriented) to conservation and protection (resource oriented), Chinese government has initiated an ambitious water reform "Building a water-saving society", which aims to achieve "harmonious coexistence between man and nature" (Wang, 2006). Our study showed that surface water use slightly decreased from 178 billion m$^3$ in 1990s to 177.6 billion m$^3$ in 2000s, with 10.7 billion m$^3$ decrease in humid basins, 9.2

billion m$^3$ increase in semi-humid basins, 270 million m$^3$ increase in semi-arid basins and 870 million m$^3$ increase in arid basins. Noteworthy, the decrease in surface water use mainly occurred in downstream regions in most basins expect Songhua and Yangtze River Basins, while surface water use continuously increased in upstream regions at a slower rate. The theory of water rights and water markets was viewed as a fundamental policy regime in the water reform. For example, the water deal between Dongyang and Yiwu counties in Zhejiang Province in 2000, the water rights trading in Zhangye city in 2002

and water allocation in Yellow River Basin in 2000s (Jiang, 2017). However, the statistical data starting in 2000 still suggested an increase of total water use and water stress between 2000 and 2010 across China (Wang et al., 2017; China Water Resources Bulletin, 2000-2010).

The story in post-2000s looks encouraging. A strict water resources management strategy – three redlines – was implemented in 2012 and statistical data showed that it began to show a slight decrease in total water use over China and

each basin (China Water Resources Bulletin, 2011-2016). Our future study will keep tracing the changes of water use and water availability and their links with water policy.

## 4.3 The lessons and experiences from China's water governance

The section 4.2 explained the lagging of water governance behind water crisis. Hence we would like to raise a question: What is the most suitable water governance for each region?

There are two different policies adopted to relieve water scarcity across the vast water-scarce northern China: Water allocation based on water rights and transboundary water transfer. The former policy is currently being applied in northwestern catchments including Shiyang, Hei and Tarim River Basins. Meanwhile, the latter policy is mainly being applied in Hai River basin, which is the destination of famous "South-to-North Water Transfer" project. The two policies are being combined in Yellow River Basin to relieve its water scarcity.

This study suggested that appropriate/optimized water allocation should be adopted in regions with high WTA and FI, while physical water transfer should be applied in regions with high WTA and low FI. For the situation of high WTA and FI, the main problem is that the imbalanced increase of water use in up and middle reaches leads to the consequent terminal lake vanishing, vegetation death, and desertification in downstream regions. Moreover, considering that the upstream complex terrains would increase the difficulties of construction of water projects, it is appropriate to adopt water allocation accompanied with water right and water price for solving environmental problems in lower reaches.

For the situation of high WTA and low FI, water allocation is not feasible here because water scarcity happens everywhere. If more surface water is forced to be released to downstream, the upstream regions will face more severe water resource shortages and consequent environmental deterioration. For example, Shanxi province, the upstream province of Hebei, Beijing and Tianjin, hasn't had enough surface water to satisfy their own demand since 1970s (Sun et al., 2016). Consequently, the development of Shanxi province heavily relied on groundwater at amount of 3.6 billion $m^3$, or 64% of total water use, in 2004 (National Bureau of Statistics of China, 2004). The excessive exploitation of groundwater has resulted in a series of environmental and geological problems, such as land subsidence, earth fissures, and a great reduction of river water flow downstream (Sun et al., 2016). Moreover, considering the higher economic value per unit water in downstream regions, for instance 15.6 and 58.4 $m^3/10^4$ GDP in 2016 in Beijing and Shanxi, respectively (National Bureau of Statistics of China, 2016), the increase of alternative water supply is a more feasible policy, including water recycling, transbasin water transfer and brackish water/sea water desalinization.

Overall, the formulation of water governance policies is challenging. The quantitative analysis of past trajectories of water scarcity in upstream, middle stream and downstream provides a sound basis for developing and implementing water governance in China.

## 5 Conclusions

The unconstrained water use in the upstream of a river basin has led to negative impacts on economy, society, and ecosystems in downstream regions. However, the quantified relationship of upstream water use and downstream water scarcity still remains unclear in China due to lack of long-term water use data. By comparing observed runoff (1970s to 2000s) and

reconstructed theoretical runoff, we analyse the trajectories of surface water use and per capita surface water availability in upstream, middle stream and downstream of China's major river basins. The scenario analysis further quantifies the impact of upstream water use on downstream water scarcity. Finally, the contribution analysis is used to identify the main drivers of water scarcity changes. Our results show that some river basins in China have experienced a dramatic increase of WTA stress

from 1970s to 2000s due to the rapid increase of water use, which mainly occurs in northern basins. In 2000s, the increase of upstream WTA stress and the decrease of downstream WTA stress occurs simultaneously, which is probably caused by the increasing upstream water use and the consequent decrease of downstream water availability outpaced by the decrease of downstream surface water use. The influence of upstream water use on downstream water scarcity is less than 10% for humid and semi-humid basins, while it is quite large for semi-arid and arid basins with WTA-impact increase from 10% in 1970s to 37%

in 2000s and FI-impact increase from 22% in 1980s to 37% in 2000s. The contribution analysis shows that the WTA contribution greatly increases in 2000s mainly in humid and semi-humid basins, while it decreases mainly in semi-arid and arid basins. The trajectories of China's water scarcity are closely related to the socioeconomic development and water policy, which thus provides valuable lessons and experiences for global water management.

**Acknowledgements**

We sincerely thank Professor Keith Richards from Cambridge University for the constructive feedback given on the study. We appreciate the efforts of three anonymous reviewers whose valuable comments helped us improve the quality of manuscript. This work was supported by the National Natural Science Foundation of China under Grant 41671021. Yongqiang Zhang acknowledges the support by CAS Pioneer Hundred Talents Program.

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

**Figure captions**

**Figure 1** The locations of the 12 basins and 37 hydrologic stations. Upstream, middle stream and Downstream were identified by red, green and yellow, respectively.

**Figure 2** The comparison of observed and natural annual discharge at the outlet stations in 12 basins. Natural discharge in most of the basins was measured at the outlet stations, but that in four basins (Hai, Shiyang, Hei and Tarim) was the discharge in the upstream tributaries because of negligible runoff generated in the downstream regions.

**Figure 3** The comparison of observed and natural annual discharge (log10 transformed) in upper, middle and lower reaches. Black triangles indicate 1960s, black dots indicate 1970s, dark grey dots indicate 1980s, light grey dots indicate 1990s and white dots indicate 2000s.

**Figure 4** Changes of WTA and FI in 12 basins and China from 1970s/1980s to 2000s. YZ, XI, MIN, QT, SH, HUA, YL, LIA, HAI, HEI, SY, TA and NA represent Yangtze, Xi (Pearl), Min, Qiantang, Songhua, Huai, Yellow, Liao, Hai, Hei, Shiyang, Tarim River Basins and National range, respectively.

**Figure 5** Trajectories of WTA and FI in upstream, middle stream and downstream regions for 9 large basins from 1970s/1980s to 2000s.

**Figure 6** Quantitative impact of upstream water use on downstream WTA (a) and FI (b). S1 is a scenario that downstream WTA (FI) is contributed by upstream water use while S2 is a scenario that downstream WTA (FI) is not contributed by upstream water use. S1-S2 (S2-S1) indicates the difference in stress between the two scenarios. YZ represents Yangtze River

Basin, XI represents Pearl River Basin, MIN represents Min River Basin, SH represents Songhua River Basin, YL represents Yellow River Basin, HAI represents Hai River Basin, SY represents Shiyang River Basin, HEI represents Hei River Basin, and TR represents Tarim River Basin. Asteroid sign indicates that the values are enlarged by 10 times for Hai River Basin to make them visible for comparison purpose.

5 **Figure 7** The combined analysis of WTA and FI showing the water scarcity trajectories in 9 river basins in the period of 1980s - 2000s. YZ represents Yangtze River Basin, XI represents Pearl River Basin, MIN represents Min River Basin, SH represents Songhua River Basin, YL represents Yellow River Basin, HAI represents Hai River Basin, SY represents Shiyang River Basin, HEI represents Hei River Basin, and TR represents Tarim River Basin.

**Figure 8** The WTA contribution in water scarcity trajectories for whole basin, upstream and downstream between different 10 periods.

**Table1: Sources of hydrological data from office and published literatures.**

| Basin (Climate) | Hydrologic station (lat/lon) | Reference |
|---|---|---|
| Pearl (humid) | Liuzhou[u](24.53/109.11), Qianjiang[u](23.68/109.1), Nanning[u](22.82/108.19), Gaoyao[d](23.26/112.22) | Chinese river sediment Bulletin (2002-2010); Dai et al., 2007a,b |
| Min (humid) | Qilijie[u](27.01/118.29), Yangkou[u](26.77/117.97), Shaxian[u](26.4/117.83), Zhuqi[d](26.12/119.15) | Chinese river sediment Bulletin (2002-2010); Dai et al., 2007a |
| Qiantang (humid) | Huashan(29.62/120.83), Zhuji(29.72/120.23), Quxian(28.98/118.87) | Chinese river sediment Bulletin (2002-2010) |
| Yangtze (semi-humid) | Yichang[u](30.75/111.3), Hankou[m](30.58/114.28), Datong[d](30.77/117.6) | Changjiang Sediment Bulletin (2010); Chinese river sediment Bulletin (2002-2010) |
| Huai (semi-humid) | Bengbu(32.74/117.23) | Chinese river sediment Bulletin (2002-2010); Pan et al., 2013; Dai et al., 2007a |
| Songhua (semi-humid) | Harbin[u](45.8/126.67), Jiamusi[d](46.83/130.13) | Chinese river sediment Bulletin (2002-2010); Tu et al., 2012; Song et al., 2009 |
| Yellow (semi-arid) | Toudaoguai[u](39.25/106.78),Huayuankou[m](34.9/113.66), Lijin[d](37.5/118.25) | Yellow River Sediment Bulletin (2000-2010); Chinese river sediment Bulletin (2002-2010) |
| Liao (semi-arid) | Tieling(42.14/122.48) | Chinese river sediment Bulletin (2002-2010); Zhang et al., 2014; Dai et al., 2007a |
| Hai (semi-arid) | Zhangjiafen[u](40.62/116.78), Xiangshuibao[u](40.51/115.18), Xiahui[u](40.62/117.17), shixiali[u](40.25/114.73) | Official source, Chinese river sediment Bulletin (2002-2010) |
| | Daomaguan[u](39.08/114.63), Xiaojue[u](38.38/113.72), Pingshan[u](38.25/114.17) | Official source |
| | Haihezha[d](39.02/117.73) | Chinese river sediment Bulletin (2002-2010); Dai et al., 2007a; Wei et al., 2016 |
| Hei (arid) | Yingluoxia[u](38.8/100.17), Zhengyixia[m](39.82/99.45) | Chinese river sediment Bulletin (2002-2010); Niu et al., 2011 |
| | Langxinshan[d](41.03/100.32) | Niu et al., 2011; Ren et al., 2015 |
| Shiyang (arid) | Caiqi[u](38.21/102.75), Hongyashan[d](38.41/102.9) | Official source |
| Tarim (arid) | Alar[u](40.5/80.99) | Chinese river sediment Bulletin (2002-2010); Zhao et al., 2010; Yang and He, 2003 |
| | Yingbazha[m](41.17/84.22), Qiala[d](40.97/86.7) | Zhao et al., 2010; Yang and He, 2003 |

Superscript: [u] represents upstream gages, [m] represents middle stream gages, and [l] represents downstream gages.

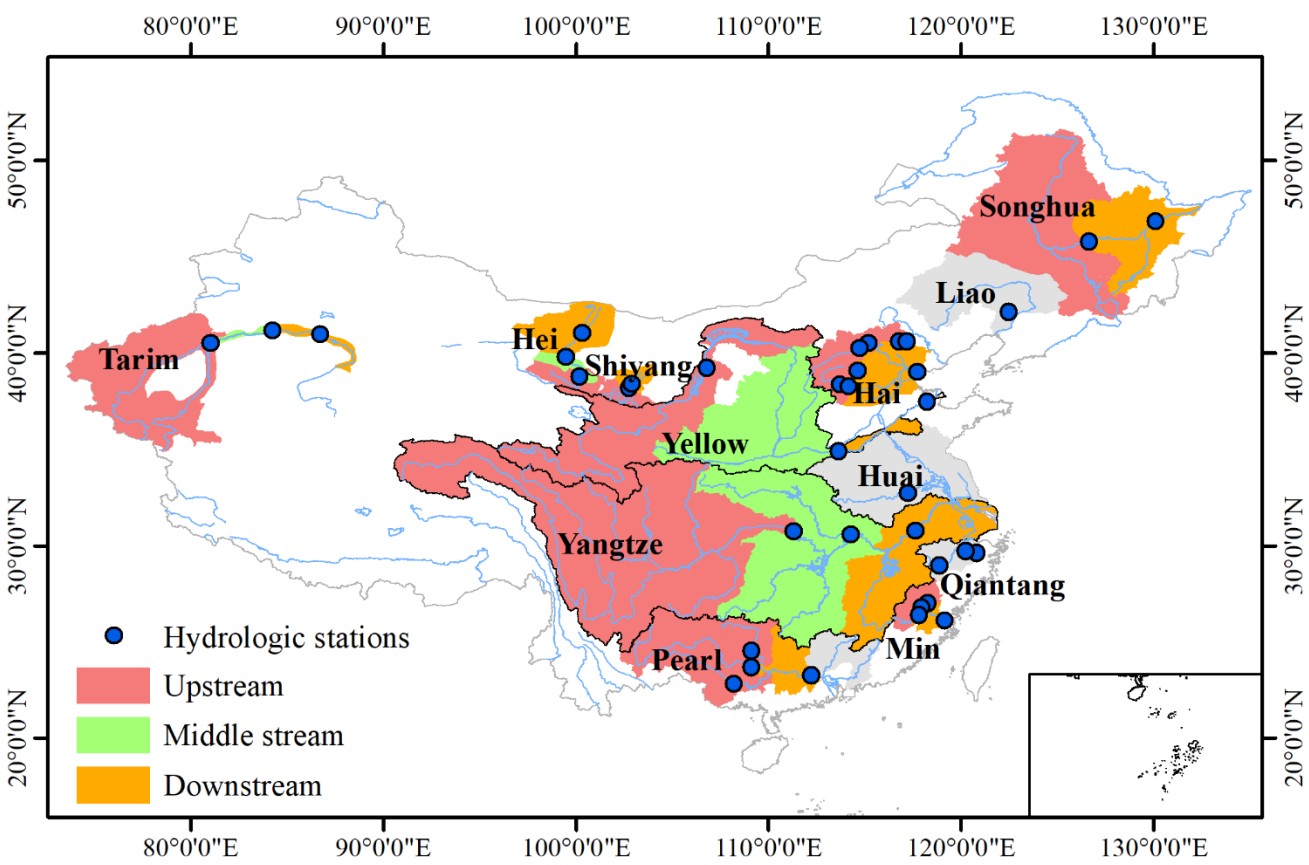

Figure 1: The locations of the 12 basins and 37 hydrologic stations. Upstream, middle stream and Downstream were identified by red, green and yellow, respectively.

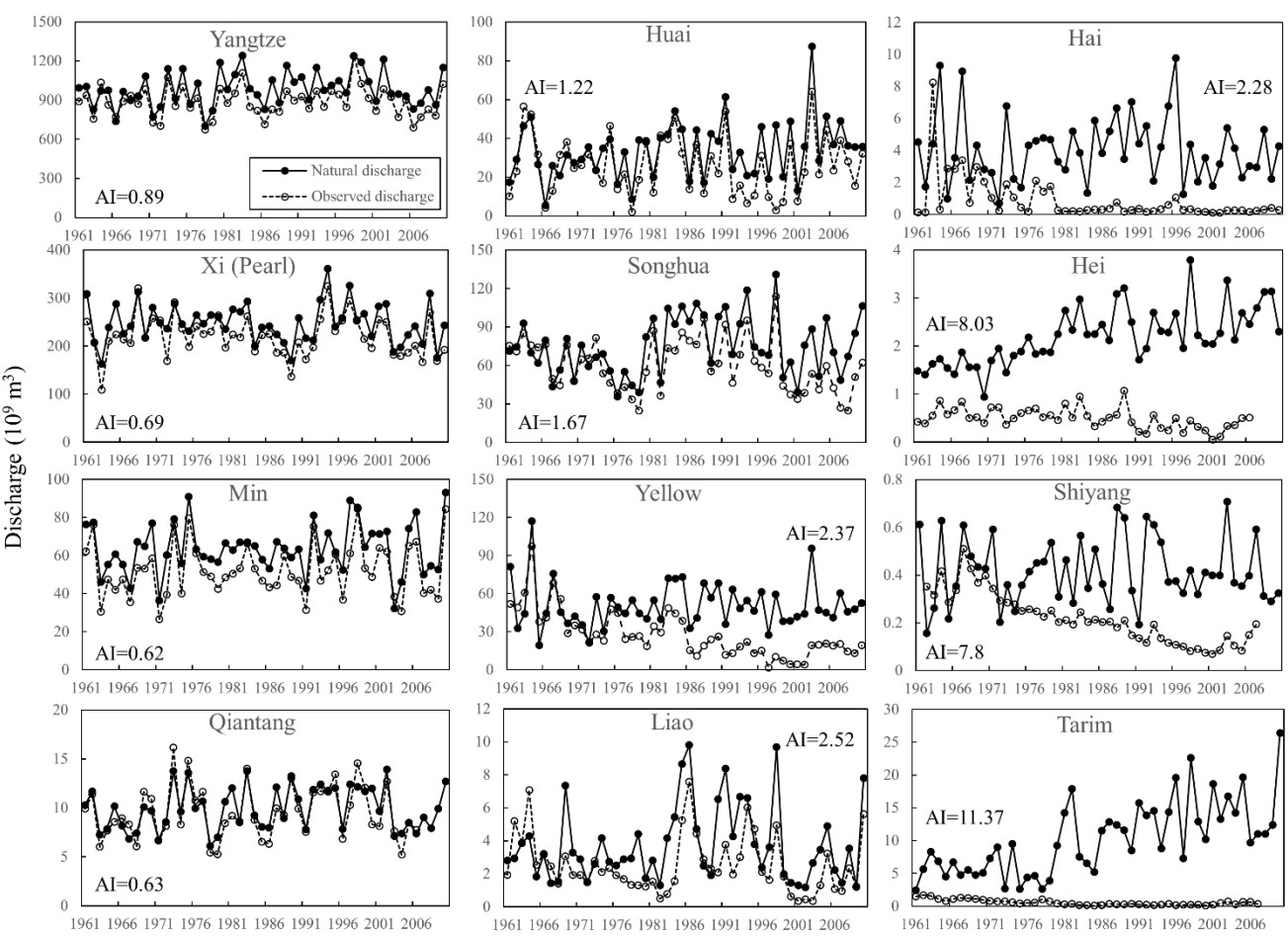

**Figure 2: The comparison of observed and natural annual discharge at the outlet stations in 12 basins. Natural discharge in most of the basins was measured at the outlet stations, but that in four basins (Hai, Shiyang, Hei and Tarim) was the discharge in the upstream tributaries because of negligible runoff generated in the downstream regions.**

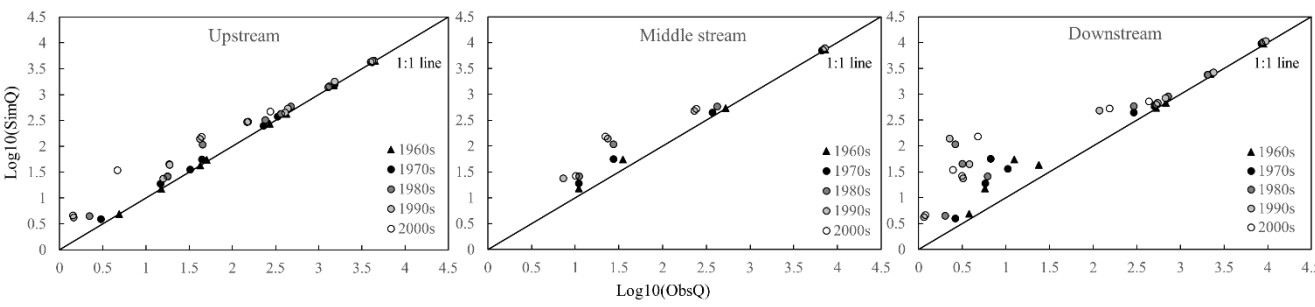

**Figure 3: The comparison of observed and natural annual discharge (log10 transformed) in upper, middle and lower reaches. Black triangles indicate 1960s, black dots indicate 1970s, dark grey dots indicate 1980s, light grey dots indicate 1990s and white dots indicate 2000s.**

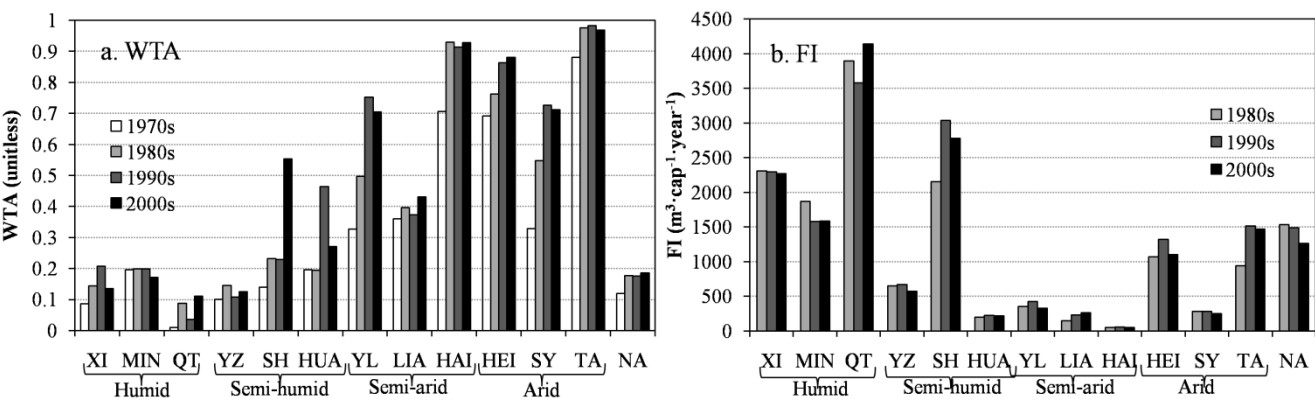

**Figure 4: Changes of WTA and FI in 12 basins and China from 1970s/1980s to 2000s. YZ, XI, MIN, QT, SH, HUA, YL, LIA, HAI, HEI, SY, TA and NA represent Yangtze, Xi (Pearl), Min, Qiantang, Songhua, Huai, Yellow, Liao, Hai, Hei, Shiyang, Tarim River Basins and National range, respectively.**

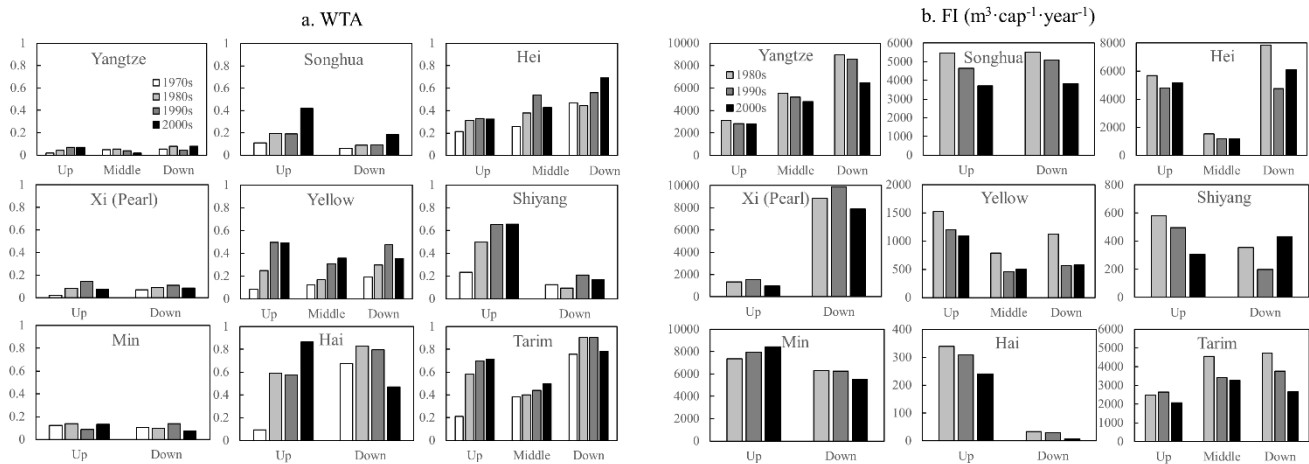

**Figure 5: Trajectories of WTA and FI in upstream, middle stream and downstream regions for 9 large basins from 1970s/1980s to 2000s.**

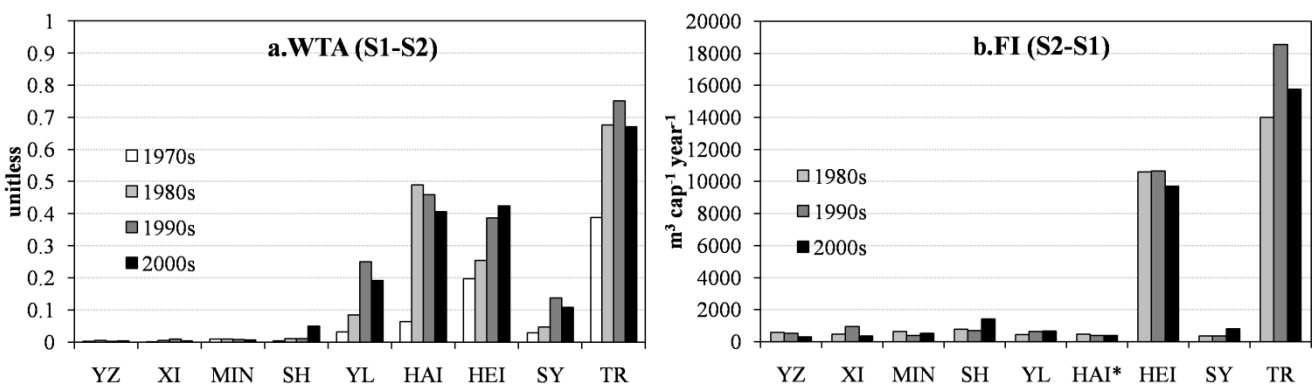

**Figure 6: Quantitative impact of upstream water use on downstream WTA (a) and FI (b). S1 is a scenario that downstream WTA (FI) is contributed by upstream water use while S2 is a scenario that downstream WTA (FI) is not contributed by upstream water use. S1-S2 (S2-S1) indicates the difference in stress between the two scenarios. YZ represents Yangtze River Basin, XI represents Pearl River Basin, MIN represents Min River Basin, SH represents Songhua River Basin, YL represents Yellow River Basin, HAI represents Hai River Basin, SY represents Shiyang River Basin, HEI represents Hei River Basin, and TR represents Tarim River Basin. Asteroid sign indicates that the values are enlarged by 10 times for Hai River Basin to make them visible for comparison purpose.**

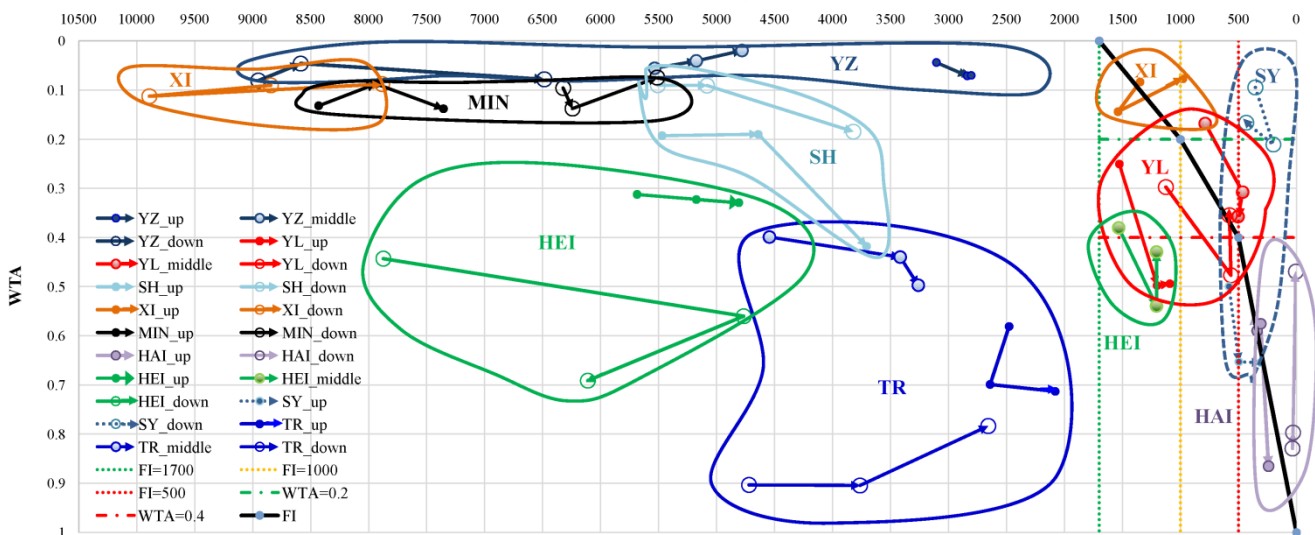

**Figure 7: The combined analysis of WTA and FI showing the water scarcity trajectories in 9 river basins in the period of 1980s - 2000s. YZ represents Yangtze River Basin, XI represents Pearl River Basin, MIN represents Min River Basin, SH represents Songhua River Basin, YL represents Yellow River Basin, HAI represents Hai River Basin, SY represents Shiyang River Basin, HEI represents Hei River Basin, and TR represents Tarim River Basin.**

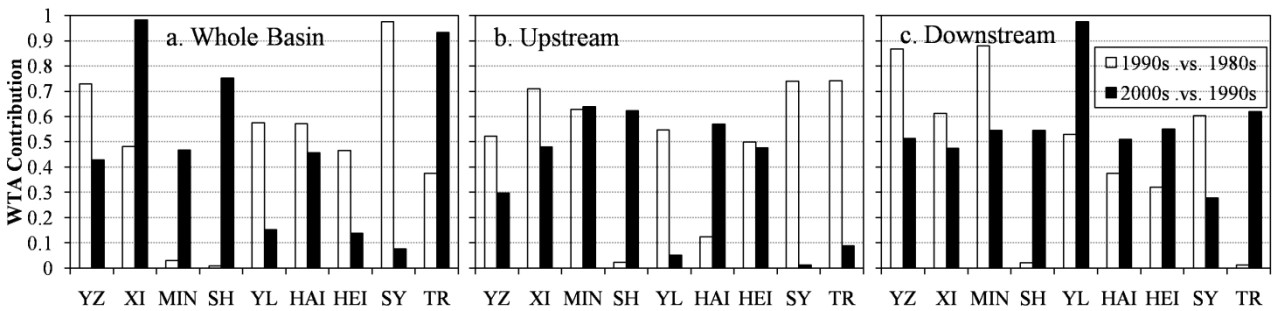

**Figure 8: The WTA contribution in water scarcity trajectories for whole basin, upstream and downstream between different periods.**