# Peer review of "Reconstructed natural runoff helps quantifying the relationship between upstream water use and downstream water scarcity in China's river basins"

_Hydrology and Earth System Sciences, 2018_

## Referee Comment (RC1) · Anonymous Referee #1 · 8 Aug 2018

The study used a well-known framework to analyze the water scarcity in some large basins in China. Although the method is not new, the topic is interesting. However, some details about the method should be added (please see the following point-to-point remarks), and the presentation of the results should be improved. In the results part, I found that the analysis was not complete for each basin, the results were not well organized, and the figures are hard to follow. These limitations made me a little bit hard to understand the results and conclusions (some are due to a lack of quantitative analysis, and some are due to a lack of complete summary and necessary discussions; particularly, the result about water scarcity was not well interpreted). Finally, the authors had three objectives, but the imbalance between upstream and downstream regions was not well quantified, and the third one was only discussed in a very simple way.

The details are given as follows:

P4L1: how did you do the model calibration to show that theta is most sensitive to topography? The details about the model calibration were missing. The theta value was constant for all the basins?

P4L2: the uncertainty of the model should be evaluated more completely. 6.9% was only the average. However, how about the spatial distribution of the uncertainty? Which basins had the largest uncertainty?

P4L3: please give references to show this framework can be suitable for annual studies. In my experience, this frame is only suitable for mean annual studies.

P4L17, here, why was ET0 calculated by the Hargreaves equation rather than the Penman equation? The gridded meteorological data can be also obtained by interpolating the station-based data to grids.

P5L2, please give the reference for the classification method of AI.

P5L5, according to Figs. 4 to 9, I think you focused more on the changes, so maybe the trend was less important. Please consider to delete the trend analysis contents to make the results more coherent.

P5L9, the definition of water scarcity is expressed by two indicators, but this is not very easy to follow, especially in Fig 9. I suggest define a new indicator, e.g., WS=WTA/Shortage? Maybe it is easier to compare this indicator among different decades, basins, and reaches.

P6L15, the correlation coefficient of natural and observed runoff means what? As defined by the authors, natural runoff and observed runoff could be totally unrelated, so I don't know what R means. 1961–1970 was the period for model calibration, so why did you show the degree of suitability of the model during 1961-2010? If the authors assumed that period from 1961–1970 was nearly natural, you should divide the period into two sub-periods: one for calibration and the rest one for validation. I noticed that the model's performance in some basins listed in the right-most column of Figure 1 was very poor during 1961–1970. Is the framework suitable for these basins?

P6L24, it is very difficult to see which gauges are in the upstream and which gauges are in the downstream. The authors should think about how to present the locations of the gauges clearly.

P6L25, can you explain why a gauge with a positive trend in rainfall can have a negative change (Fig. 3)?

P6L29, in northwest of China, such as Heihe, Tarim, river runoff is mostly contributed by snow melt. Is the framework suitable for these basins?

P6L25, P71, the authors gave subjectively the reasons for the trend (a significant increase in rainfall, recent global warming). I don't see any supporting analysis.

P7L11-15, from Figure 4, I can't see these interesting analyzes. And, please add the AI in this figure.

P7L14-15, this is also too subjective.

P7L20, in Figure 5a, I suggest add an average of 1970s~2000s for each basin. Here, how did you define "continuously"? Obviously, WTA in the Yangtze, Pearl, Min River and Songhua did not increase monotonously.

P7L17-25, these results should be discussed to give the possible reasons.

P8L5, Figure 7 is about water shortage, so I don't know why the authors were talking about surface water availability.

P8L11, water availability is determined by natural runoff, so I can't understand why population can affect water availability.

P8L19, from Figure 9a, I can't see the aggravation of water scarcity in China. This figure is not visual to show this aggravation trend.

P8L25, water scarcity is defined with water stress and water shortage, here, why is it related to surface water availability?

P8L28, fig. 9a and 9c cannot show this competition (at least I don't know how to interpret). And this paragraph was about water scarcity, but the authors were talking about water withdrawal. So it is very hard to understand these sentences.

In Figure 8: in the Liao, Huai, and Qiantang, why were there no upstream, middle, and downstream?

P9L4-5, no analysis supporting the statement here.

P9L16, the possible impacts of the policies on water scarcity in all the basins were not fully discussed.

---

## Referee Comment (RC2) · Anonymous Referee #2 · 31 Aug 2018

Reconstructed natural runoff suggests imbalance in water scarcity between upstream and downstream regions of China's river basins Here, the authors presented a framework for quantifying the change in water scarcity at major river basins of China. Although, the study is interesting the methodology is not new and the manuscript is poorly written. For publishing purpose, the entire manuscript should be presented in a high quality format. The details of methodology is also not clear. In addition, the authors did not provide equal importance for all the objectives mentioned in the study. I am including the comments here, Introduction The introduction should be improved with proper citations and sentences which shows the importance of the current study. Page 2 second paragraph is confusing. The sentences should be clear. Please add

references for "A recent study has shown that the impact of anthropologic interventions on water scarcity is not always negative". Line 21 – 22 (page 2) is confusing. Please correct the sentence. Line 26 is not clear. Please rewrite the entire paragraph. Page 3. The presentation of objectives is poor and not clear. Please write with specific reasoning. In addition, the sentence "The answers will provide experiences and lessons for global water resources management" is not matching here. Overall, the introduction is too short and not clearly written. Please provide more information on the importance of water scarcity analysis by using different indices such as, water stress and water shortage. Please try to link the importance of Fu-Budyko in water scarcity analysis in a river basin scale. Materials and methods The manuscript needs more explanation on methodology section. Starting the paragraph with 'because' is not recommended. In table 1 provide the lat/lon for gage locations. Need more explanation on the section Hydrological data reliability. Line 29 – please replace e.g. by such as. Page 4. The sentence "The steeper the catchment, the smaller was the parameter" is not clear. Need more explanation on the catchment parameter (theta) used in the study? The equation 1 shows the Fu – Budyko frame work, and it is a function of aridity index. But the authors did not mention it here. But in page 5 authors introduced the AI (aridity index). It will make confusion to the readers. Please rewrite the section accordingly. Expand the unit mm/a (line 13). Hargreaves is not a suitable method for quantifying the potential ET hence it is only based on Tmax and Tmin. Please mention the drawback in the manuscript. What does the value 17.8 indicates in the equation 2. Be more specific. The line 23 – 27 is not clear. Please rewrite. Page 5. Line 2 – 3. What is the basis of this classifications? Include references. The trend analysis section is not clear. Need more explanation including the equations used. Line 12 – 13 is not clear. Rewrite. Line 17 Populations or population Line 10 – 19 please rewrite. The definition of WW is confusing. Please explain the Qnat and Qobs more specifically. Page 6. Is it population count data or population data? Please rewrite the section 'water stress and shortage'. Overall, the methodology section is not clearly written and confusing for the readers. Please improve the section.

Results The first sentence is not clear. What does the term sustainability indicates. Why did the authors calculate the correlation between observed and natural runoff? Need a clear explanation for this section. Page 6. The line 15 – 18 is not written well. Please improve the writing quality. Page 7. Line 18 shows that the authors selected only 9 large river basins for analysis. Please explain the reasons. The explanation for the questions "How did the imbalance in surface water scarcity develop between upstream and downstream regions? and What do we learn from China's water management strategies?" are not sufficient in the manuscript. Explain how the model framework is performing for different regions such as, snow regions in the manuscript. The discussion on percentage decrease in surface water withdrawal is not clear. Please explain the possible reasons. Page 9. Line 26 – 29 is not clear. The discussion section is not sufficient and well written.

Please also note the supplement to this comment:
https://www.hydrol-earth-syst-sci-discuss.net/hess-2018-364/hess-2018-364-RC2-supplement.pdf

---

## Author Comment (AC1) · 18 Oct 2018

General comments The study used a well-known framework to analyze the water scarcity in some large basins in China. Although the method is not new, the topic is interesting. However, some details about the method should be added (please see the following point-to point remarks), and the presentation of the results should be improved. In the results part, I found that the analysis was not complete for each basin, the results were not well organized, and the figures are hard to follow. These limitations made me a little bit hard to understand the results and conclusions (some are due to a lack of quantitative analysis, and some are due to a lack of complete summary and

necessary discussions; particularly, the result about water scarcity was not well interpreted). Finally, the authors had three objectives, but the imbalance between upstream and downstream regions was not well quantified, and the third one was only discussed in a very simple way. Response: We thank you for your recognition of our work and appreciate the constructive comments and insightful suggestions that will help improve our paper. We will address your concerns in the revisions. The detailed responses for your point-to-point remarks are listed below.

Specific comments: P4L1: how did you do the model calibration to show that theta is most sensitive to topography? The details about the model calibration were missing. The theta value was constant for all the basins? Response: The parameter theta of Fu-Budyko framework does change from one basin to another and from upstream to downstream. Here three basins, Hei River, Yellow River and Yangtze River, were taken as examples to show the change of theta in the following table. River Pup PETup thetaup Pdown PETdown thetadown Yangtze River 705 1083 1.7 1106 996 2.0 Yellow River 498 866 1.7 433 1027 2.3 Hei River 375 916 1.3 215 1027 2.0 The climate conditions of the three basins are different, from humid Yangtze River basin, to semi-arid Yellow River basin and arid Hei River basin. Some have more arid upstream regions while others have more arid downstream regions. The similarity, however, was identified that the theta of upstream is lower when compared to its downstream counterpart. Given the fact that upstream regions have steeper terrains, the lower theta is probably related to the topography. The result is consistent with study from Sun et al. (2007), indicating that three factors - infiltration rate, water storage capacity and average slope - had impact on the parameter theta of Fu-Budyko framework.

Reference: Sun, F., Yang, D., Liu, Z., and Cong, Z., 2007. Study on coupled water-energy balance in Yellow River basin based on Budyko Hypothesis (in Chinese). Journal of Hydraulic Engineering, 38(4), 409-416

P4L2: the uncertainty of the model should be evaluated more completely. 6.9% was only the average. However, how about the spatial distribution of the uncertainty? Which

basins had the largest uncertainty? Response: This is a very good point. Generally, the biases are smaller in humid basins while larger in arid basins. The southeastern basins (Min and Qiantang) have the lowest biases around 3%. Followed are the southern basins (Pearl and Yangtze) and northeastern basins (Songhua) with biases around 6%. Yellow basin and the northwestern basins have approximately 10% biases. Hai river basin has the highest biases of ∼20%.

P4L3: please give references to show this framework can be suitable for annual studies. In my experience, this frame is only suitable for mean annual studies. Response: Zhang et al. (2008) has tested the Budyko model over 265 Australian catchments at different time scales, including mean annual, annual, monthly and daily. They found at annual scale, the model works well for most of the catchments with 90% of them having values of the coefficient of efficiency greater than 0.5 and less than 3% of the catchments have bias values greater than 10%. Meanwhile, there are some catchments where the model performed poorly. A reference will be added.

Reference: Zhang, L., Potter, N., Hickel, K., Zhang, Y., and Shao, Q., 2008. Water balance modeling over variable time scales based on the Budyko framework - Model development and testing. Journal of Hydrology, 360, 117-131.

P4L17: here, why was ET0 calculated by the Hargreaves equation rather than the Penman equation? The gridded meteorological data can be also obtained by interpolating the station-based data to grids. Response: Both equations were used in the study. The Hargreaves method was chosen because only temperature and precipitation were available in the gridded meteorological dataset. And the PM-based potential ET from pointed dataset was used to corrected the Hargreaves-based potential ET, which will be greatly improve the accuracy especially in the eastern regions. Both the gridded and pointed meteorological data have their advantages and disadvantages: the pointed dataset contains more meteorological variables but is sparse in the western regions, while the gridded dataset is denser in the western regions but contains only temperature and precipitation. Considering the complex terrain and dry climate in western

regions, we think the distribution of meteorological gauges is more influential factor for the accuracy of interpolation of potential ET. Thus the combination of gridded data and pointed data was chosen in this study to reduce the errors of potential ET in western regions. Additional notes will be added in revision for clarification.

P5L2, please give the reference for the classification method of AI. Response: The classification of AI is following the method of Ponce et al. (2000) with arid, semi-arid, semi-humid, and humid regions ranging from 12~5, 5~2, 2~0.75, and 0.75~0.375. In this manuscript, there were mistakes to label the limits of AI and these will be corrected in the revised manuscript. The following references will be added: Arora, V.K., 2002. The use of the aridity index to assess climate change effect on annual runoff. Journal of Hydrology, 265(1-4), 164-177. Ponce, V.M., Pandey, R..P., and Ercan, S., 2000. Characterization of drought across climate spectrum. Journal of Hydrologic Engineering, ASCE, 5(2), 222-224.

P5L5, according to Figs. 4 to 9, I think you focused more on the changes, so maybe the trend was less important. Please consider to delete the trend analysis contents to make the results more coherent. Response: Thank you for your suggestion. The reason for keeping the trend analysis in the manuscript was in case someone is interested in the trends. After careful consideration of your suggestion, we decide to delete Fig. 3 which demonstrates the P and potential ET trends. Meanwhile, the Fig. 2 is still kept because we think the trends of natural and observed runoff are helpful for readers to understand the changes of water scarcity.

P5L9, the definition of water scarcity is expressed by two indicators, but this is not very easy to follow, especially in Fig. 9. I suggest define a new indicator, e.g., WS=WTA/Shortage? Maybe it is easier to compare this indicator among different decades, basins, and reaches. Response: Thank you for your thoughts. In fact, "Shortage" refers to the available surface water resources per capita and it is related to the demographic-driven change of water scarcity. We will change the description of "Shortage" and use a new abbreviation instead of "Shortage".
P6L15, the correlation coefficient of natural and observed runoff means what? As defined by the authors, natural runoff and observed runoff could be totally unrelated, so I don't know what R means. 1961-1970 was the period for model calibration, so why did you show the degree of suitability of the model during 1961-2010? If the authors assumed that period from 1961-1970 was nearly natural, you should divide the period into two sub-periods: one for calibration and the rest one for validation. I noticed that the model's performance in some basins listed in the right most column of Figure 1 was very poor during 1961-1970. Is the framework suitable for these basins? Response: At the beginning, we thought the correlation between observed and natural runoff might reflect the human interventions on runoff. We will reevaluate our earlier conclusion. We will use correlation coefficient in calibration period to show the model performance here. And we worried about if five-years would be long enough to calibrate the model. We will try the suggestion to divide 1961-1970 into calibrate and validate period. For the arid basins, we divided the arid basins into upper mountainous sub-basins and downstream sub-basins, and applied Budyko framework in the former sub-basins. For the downstream sub-basins, we use the observed runoff and evapotranspiration to calculate runoff. Please see the answer to P6L29 for the detailed explanation.

P6L24, it is very difficult to see which gauges are in the upstream and which gauges are in the downstream. The authors should think about how to present the locations of the gauges clearly. Response: Thank you for the comments. We will label these hydrological gauges in Figure 1 to make these visible.

P6L25, can you explain why a gauge with a positive trend in rainfall can have a negative change (Fig. 3)? Response: Here the rainfall's change was not calculated as the trend (mm/year) * year, instead it was calculated as the differences between two periods - 2000s and 1960s. Thus the change of a gauge was only dependent on the differences between 2000s and 1960s but not the trend. For those gauges with fluctuations and no significant trend, it is possible that a gauge with a positive trend has a negative change.

P6L29, in northwest of China, such as Heihe, Tarim, river runoff is mostly contributed

by snow melt. Is the framework suitable for these basin? Response: Budyko framework performs bad in arid and snow basins, which has been proved in the previous studies. Here we found that Du et al. (2016) successfully applied a Budyko framework in arid Heihe River Basin by dividing it into six sub-basins according to basin characteristics. They calibrated the model separately in different sub-basins and found the model performed quite well in the upper mountainous regions with little interference of human activities. So we also divided the arid basins into upper mountainous sub-basins and downstream sub-basins, and applied Budyko framework in the former sub-basins. For the downstream sub-basins, we use observed runoff and evapotranspiration to calculate runoff. Detailed explanation will be added in method section.

Reference: Du, C., Sun, F., Yu, J., Liu, X., and Chen, Y., 2016. New interpretation of the role of water balance in an extended Budyko hypothesis in arid regions. Hydrology and Earth System Sciences, 20, 393-409.

P6L25, P71, the authors gave subjectively the reasons for the trend (a significant increase in rainfall, recent global warming), I don't see any supporting analysis. Response: There are two reasons for the increase of observed runoff, one for increasing precipitation and the other for decreasing evapotranspiration. Given the insignificant change of potential evapotranspiration, we think the main driver for increasing runoff is the increase of precipitation.

P7L11-15, from Figure 4, I can't see these interesting analyzes. And, please add the AI in this figure. Response: Thank you for the suggestion on Figure 4. We think the statement of the paragraph (P7L11-15) is too subjective. Thus we will use numbers to describe the change of water stress and include additional discussions here.

P7L14-15, this is also too subjective. Response: As mentioned above, we are considering to describe the change of water stress using numbers here.

P7L20, in Figure 5a, I suggest add an average of 1970s∼2000s for each basin. Here, how did you define "continuously"? Obviously, WTA in the Yangtze, Pearl, Min River

and Songhua did not increase monotonously. Response: Thank you for the suggestion on Figure 5. And we are sorry for the improper "continuously". Because the fluctuations of WTA are small in these humid basins within a range less than 2%, so we think the fluctuations might be caused by the modelling errors and can be ignored. We will fix the improper word and replace it with a new one.

P7L17-25, these results should be discussed to give the possible reasons. Response: Thank you for the helpful suggestion. We will further discuss the turning point of the decreasing surface water stress for different basins in the revised manuscript, which is supposed to be related to the economic development.

P8L5, Figure 7 is about water shortage, so I don't know why the authors were talking about surface water availability. Response: "Shortage" is defined as surface water availability per capita, so both water availability and population can influence water shortage. In this paragraph, we aimed to explain the converse phenomenon of water shortage between northern and southern basins, which was mainly related to the surface water availability. We will add an explanation before the paragraph and change the description of "Shortage" in the revised manuscript.

P8L11, water availability is determined by natural runoff, so I can't understand why population can affect water availability. Response: We feel sorry about the typing mistakes in this paragraph. It should be "water shortage" or "water availability per capita" but not "water availability". These mistakes will be corrected in the coming revised manuscript.

P8L19, from Figure 9a, I can't see the aggravation of water scarcity in China. This figure is not visual to show this aggravation trend. Response: When the dots of water scarcity move to up-right direction, the aggravation of water scarcity happened because of higher WTA and lower Shortage. We will try to seek a solution by either changing the figure or adding explanation to make the figure more understandable.

P8L25, water scarcity is defined with water stress and water shortage, here, why is

it related to surface water availability? Response: We agree that it is an improper expression here. It should be "dramatic increase of surface water withdrawal and little change of water availability per capita, suggesting it is demand-driven water scarcity in semi-humid/arid basins". We will change the expression here.

P8L28, fig. 9a and 9c cannot show this competition (at least I don't know how to interpret). And this paragraph was about water scarcity, but the authors were talking about water withdrawal. So it is very hard to understand these sentences. Response: We are sorry for the bad exhibition of Figure 9. We will try to revise the figure or add more statement to explain the figure. And water withdrawal was mentioned here because it is the most influential factor on water scarcity for northern basins. We will reorganize the paragraph to make it less confused.

In Figure 8: in the Liao, Huai, and Qiantang, why were there no upstream, middle, and downstream? Response: Because the hydrological data of some gauges is not available in the three basins. For example, the record of hydrological data in Liao's upstream gauge started from 1984, which was too short to conduct the analysis; the hydrological data of Huai's downstream gauge was missing; and there was only hydrological data in tributary gauges for Qiantang basin. A short explanation will be added in the data section.

P9L4-5, no analysis supporting the statement here. Response: We are sorry for the improper statement here. Stricter expression will replace the old one as "This study showed that climate change was the major driver of natural runoff."

P9L16, the possible impacts of the policies on water scarcity in all the basins were not fully discussed. Response: Thank you for the advise. More discussion about the polices and their impacts on relieving water scarcity in China will be added in the discussion section.

Please also note the supplement to this comment:

https://www.hydrol-earth-syst-sci-discuss.net/hess-2018-364/hess-2018-364-AC1-supplement.pdf
* * *

---

## Author Comment (AC2) · 18 Oct 2018

General comments Here, the authors presented a framework for quantifying the change in water scarcity at major river basins of China. Although, the study is interesting the methodology is not new and the manuscript is poorly written. For publishing purpose, the entire manuscript should be presented in a high quality format. The details of methodology is also not clear. In addition, the authors did not provided equal importance for all the objectives mentioned in the study. Response: We thank you for your recognition of our work, and appreciate the favorable comments and insightful suggestions that have helped improve our paper. The detailed responses are as below.

[Figure]

Specific comments: Introduction The introduction should be improved with proper citations and sentences which shows the importance of the current study. Response: Thank you for the helpful suggestion. We plan to rewrite the introduction section to focus on the competition of water resources between upstream and downstream.

Page 2 second paragraph is confusing. The sentences should be clear. Please add references for "A recent study has shown that the impact of anthropologic interventions on water scarcity is not always negative". Response: It is the paragraph that will be rewritten and extended to sum up the studies of water resources changes between upstream and downstream regions systematically. And the reference for the mentioned sentence is as following:

Reference: Veldkamp, T.I.E., Wada, Y., Aerts, J.C.J.H., Döll, P., Gosling, S.N., Liu, J., Masaki, Y., Oki, T., Ostberg, S., Pokhrel, Y., Satoh, Y., Kim, H., and Ward, P.J.: Water scarcity hotspots travel downstream due to human interventions in the 20th and 21st century, Nat. Commun., 8, 15697, doi:10.1038/ncomms15697, 2017.

Line 21-22 (page2) is confusing. Please correct the sentence. Line 26 is not clear. Please rewrite the entire paragraph. Response: This paragraph was talking about the method and why the method was chosen in this study. We will rewrite the paragraph to make it more understandable for readers.

Page 3. The presentation of objects is poor and not clear. Please write with specific reasoning. In addition, the sentence "The answers will provide experiences and lessons for global water resources management" is not matching here. Response: Thank you for the suggestion. A good presentation of objectives will make the study more logic and easy to follow. We will rewrite the objects in the revised manuscript. And the sentence mentioned above might be deleted or corrected here.

Overall, the introduction is too short and not clearly written. Please provide more information on the importance of water scarcity analysis by using different indices such as, water stress and water shortage. Please try to link the importance of Fu-Budyko

in water scarcity analysis in a river basin scale. Response: Thank you for the specific instructions. The introduction section will be thoroughly rewritten accordingly in the revised manuscript.

Materials and methods The manuscript needs more explanation on method section. Response: More explanations will be added into the method section accordingly.

Starting the paragraph with 'because' is not recommended. In table 1 provide the lat/lon for gauge locations. Need more explanation on the section Hydrological data reliability. Response: The first sentence will be corrected and lat/lon information will be added into Table 1. More information about the extracting and processing observed runoff will be provided to explain the data reliability.

Line 29 - please replace e.g. by such as. Response: This will be corrected in the revised manuscript.

Page 4. The sentence "The steeper the catchment, the smaller was the parameter" is not clear. Need more explanation on the catchment parameter (theta) used in the study? Response: During the calibration, we found that the theta of upstream is lower than that of downstream for all basins, no matter dry climate or humid climate. Given the fact that more steeper terrains in upstream, we think the topography most likely contributes to the change of theta. The observation is consistent with study from Sun et al. (2007), who thought that three factors - infiltration rate, water storage capacity and average slope - had impact on the parameter theta of Fu-Budyko framework. More explanation about the change of theta will be added here.

Reference: Sun, F., Yang, D., Liu, Z., and Cong, Z., 2007. Study on coupled water-energy balance in Yellow River basin based on Budyko Hypothesis (in Chinese). Journal of Hydraulic Engineering, 38(4), 409-416

The equation 1 shows the Fu-Budyko framework, and it is a function of aridity index. But the authors did not mention it here. But in page 5 authors introduced the AI (aridity

index). It will make confusion to the readers. Please rewrite the section accordingly. Response: Thank you for pointing out the mistake. It will be corrected in the revised manuscript.

Expand the unit mm/a (line 13) Response: It will be corrected to mm/year.

Hargreaves is not a suitable method for quantifying the potential ET hence it is only based on Tmax and Tmin. Please mention the drawback in the manuscript. Response: The Hargreaves method was chosen because only temperature and precipitation were available in the gridded meteorological dataset. And the PM-based potential ET from pointed dataset was used to corrected the Hargreaves-based potential ET, which will be greatly improve the accuracy especially in the eastern regions. More details will be added here.

What does the value 17.8 indicates in the equation 2. Be more specific. Response: The 0.0023, 17.8 and 0.5 in equation 2 are the default parameters of Hargreaves equation. The explanation of the parameters will be added in the revised manuscript.

Then line 23-27 is not clear. Please rewrite. Response: The paragraph will be rewritten in the revised manuscript.

Page 5. Line 2-3. What is the basis of this classifications? Include references. Response: The classification of AI is based on the method presented by Arora (2002) with arid, semi-arid, semi-humid, and humid regions ranging from 12~5, 5~2, 2~0.75, and 0.75~0.375. In this manuscript, there was mistakes to label the limits of AI and these will be corrected in the revised manuscript. The references are as follows: Arora, V.K., 2002. The use of the aridity index to assess climate change effect on annual runoff. Journal of Hydrology, 265(1-4), 164-177.

The trend analysis section is not clear. Need more explanation including the equations used. Response: We will consider either include the equations or add citations to make the method more understandable.

Line 12-13 is not clear. Rewrite. Line 10-19 please rewrite. Please rewrite the section 'water stress and shortage'. Response: The statements mentioned above will be written in the revised manuscript to make them easier to follow.

Line 17 Populations or population Response: Populations will be changed to population.

The definition of WW is confusing. Please explain the Qnat and Qobs more specifically. Response: Here, the WW refers to local water withdrawal which should subtract the WW in previous reach to avoid double counting. More explanation about WW, Qnat and Qobs will be added here.

Page 6. Is it population count data or population data? Response: It is population count data here.

Overall, the methodology section is not clearly written and confusing for the readers. Please improve the section. Response: We will consider all suggestions above and improve the method section to avoid confusion.

Results The first sentence is not clear. What does the term sustainability indicates. Why did the authors calculate the correlation between observed and natural runoff? Need a clear explanation for this section. Response: The suitability refers to the reliability of the model when it is applying in China. The sentence will be reorganized to make it clearer. At the beginning, we thought the correlation between observed and natural runoff might reflect the human interventions on runoff. Now we will reevaluate the conclusion based on additional analysis. We will use correlation coefficient in calibration period to show the model performance here.

Page 6. The line 15-18 is not written well. Please improve the writing quality. Response: We will reorganize the paragraph by using numbers to make it less subjective.

Page 7. Line 18 shows that the authors selected only 9 large river basins for analysis. Please explain the reasons. Response: The rest three basins missed some critical data

so can't do the upstream-downstream analysis. For example, the record of hydrological data in Liao's upstream gauge started from 1984, which was too short to conduct the analysis; the hydrological data of Huai's downstream gauge was missing; and there was only hydrological data in tributary gauges for Qiantang basin. A short explanation will be added in the data section.

The explanation for the questions "How did the imbalance in surface water scarcity develop between upstream and downstream regions? and What do we learn from China's water management strategies?" are not sufficient in the manuscript. Response: The second objective will be further discussed in the discussion section by linking with the local policies and economic development. And the last objective might be deleted because it is a little subjective and overlapped with the second objective.

Explain how the model framework is performing for different regions such as, snow regions in the manuscript. Response: Previous studies showed that the Budyko framework performs better in humid regions than in arid regions. Our study proved the result. Further explanations will be added to describe the result.

The discussion on percentage decrease in surface water withdrawal is not clear. Please explain the possible reasons. Response: As mentioned above, more discussions will be added by linking the economic development and water policies with the result in the revised manuscript.

Page 9. Line 26-29 is not clear. The discussion section is not sufficient and well written. Response: We consider to rewrite the discussion section to make it focusing on this study and the influence of local water policies and economic development on water scarcity.

Please also note the supplement to this comment:
https://www.hydrol-earth-syst-sci-discuss.net/hess-2018-364/hess-2018-364-AC2-supplement.pdf

---

## Author Response (AR1)

RESPONSE TO REVIEWER #1

General comments

The study used a well-known framework to analyze the water scarcity in some large basins in China. Although the method is not new, the topic is interesting. However, some details about the method should be added (please see the following point-to point remarks), and the presentation of the results should be improved. In the results part, I found that the analysis was not complete for each basin, the results were not well organized, and the figures are hard to follow. These limitations made me a little bit hard to understand the results and conclusions (some are due to a lack of quantitative analysis, and some are due to a lack of complete summary and necessary discussions; particularly, the result about water scarcity was not well interpreted). Finally, the authors had three objectives, but the imbalance between upstream and downstream regions was not well quantified, and the third one was only discussed in a very simple way.

Response: We appreciate reviewer #1's constructive comments which help us improve our manuscript a lot. According to the comments, we thoroughly redrafted the manuscript: a scenario analysis has been added to quantify the impact of upstream water use on downstream water scarcity, and anew water scarcity indicator was defined to explore the main driver of water scarcity. The objectives changes correspondingly. In addition, the discussion part has been rewritten and the related water policy was fully discussed. We hope the new version will satisfy the criterion of publication. The point-to-point reply is listed below.

Specific comments:

P4L1: how did you do the model calibration to show that theta is most sensitive to topography? The details about the model calibration were missing. The theta value was constant for all the basins?

Response: We realized that the model calibration was not necessary in this study. As the P, ET0 and observed runoff are known variables, the ordinary least squares method was adopted to fit the model parameter using the P, ET0 and observed runoff in the period of 1961-1970, then the fitted parameter was used to calculate the natural runoff in the following period. Only the upstream, middlestream and downstream gauge stations were kept and the irrelevant tributaries were removed. Although the parameter was recalculated in the new manuscript, the upstream regions still have lower value compared to its downstream counterpart, which is probably related to the steeper terrains in upstream regions. The explanation has been added in the P5L13-15 as "*Using the observed ET0, P and observed discharge, the parameter θ was calculated using the least-square data fitting method for the period 1961–1970, then the fitted parameter was used to reconstruct decadal natural runoff for the period 1971–2010.*"

P4L2: the uncertainty of the model should be evaluated more completely. 6.9% was only the average. However, how about the spatial distribution of the uncertainty? Which basins had the largest uncertainty?

Response: Given the auto-fitted parameters, the biases were quite small with less than 1% for humid basins (Pearl, Min and Qiantang River Basins), 1% for semi-humid basins (Yangtze, Songhua and Huai River Basins), -3% for semi-arid basins (Yellow, Hai and Liao River Basins) and 3% for the upstream of arid basins (Hei, Shiyang and Tarim River Basins).

P4L3: please give references to show this framework can be suitable for annual studies. In my experience, this frame is only suitable for mean annual studies.
Response: A reference has been added in the discussion part in P11L12-15 as "*This has been proved by Zhang et al. (2008), who has tested the Budyko model over 265 Australian catchments at different time scales, including mean annual, annual, monthly and daily. They found at annual scale, the model works well for most of the catchments with 90% of them having values of the coefficient of efficiency greater than 0.5 and less than 3% of the catchments have bias values greater than 10%.*"
P4L17: here, why was ET0 calculated by the Hargreaves equation rather than the Penman equation? The gridded meteorological data can be also obtained by interpolating the station-based data to grids.
Response: A explanation of the combined use of HG-ET0 and PM-ET0 was added in section 2.2.2 in P5L17-23 as "*Two equations – Hargreaves (HG) and Penman-Monteith (PM) – were used to estimate ET0 (Allen et al., 1998). The HG-ET0 was based on gridded dataset at monthly scale while PM-ET0 was based on pointed dataset at daily scale. The PM equation ranked as the best equation for estimating ET0 but the sparse distribution of climate stations limited its application in western China. The continuous spatial coverage of gridded dataset can provide full estimation of HG-ET0 in western China. However, large discrepancies between HG-ET0 and PM-ET0 were found in different regions over the world in previous studies (Temesgen et al., 2005; Gavilan et al., 2006; Trajkovic, 2007; Bautista et al., 2009; Sivaprakasam et al., 2011; Berti et al., 2014). Thus more accurate ET0 can be obtained by combining two estimations.*"

P5L2, please give the reference for the classification method of AI.
Response: The classification has been added in P5L6-9 as "*In this study, AI of each catchment was calculated at mean annual scale for the period of 1961-2010 and the catchments were classified into humid, semi-humid, semi-arid and arid for AI ranging from 0.375~0.75, 0.75~2, 2~5 and 5~12, respectively (Ponce et al., 2000; Arora, 2002).*"

P5L5, according to Figs. 4 to 9, I think you focused more on the changes, so maybe the trend was less important. Please consider to delete the trend analysis contents to make the results more coherent.
Response: Thank you for your suggestion. The reason for keeping the trend analysis in the manuscript was in case someone is interested in the trends. After careful consideration of your suggestion, we have deleted all the trend analysis contents to make the results coherent.

P5L9, the definition of water scarcity is expressed by two indicators, but this is not very easy to follow, especially in Fig. 9. I suggest define a new indicator, e.g., WS=WTA/Shortage? Maybe it is easier to compare this indicator among different decades, basins, and reaches.

Response: Thank you for the useful comment. We redefined a new water scarcity indicator in section 2.2.5 and the explanation of the new indicator was listed as "*Kummu et al. (2016) used per capita water use (WTA×FI) to show the variation of water scarcity. However, it probably failed to describe water scarcity variation because of the cancelling out of the two indicators: The increase of water scarcity was indicated by higher WTA but lower FI. Hence we proposed a new water scarcity definition, per capita remaining available water, which is described as (1-WTA) ×FI. The new definition of water scarcity can better show the change of water scarcity through the consistent change of (1-WAT) and FI.*"

P6L15, the correlation coefficient of natural and observed runoff means what? As defined by the authors, natural runoff and observed runoff could be totally unrelated, so I don't know what R means. 1961-1970 was the period for model calibration, so why did you show the degree of suitability of the model during 1961-2010? If the authors assumed that period from 1961-1970 was nearly natural, you should divide the period into two sub-periods: one for calibration and the rest one for validation. I noticed that the model's performance in some basins listed in the right most column of Figure 1 was very poor during 1961-1970. Is the framework suitable for these basins?

Response: In the new manuscript, the ordinary least squares method was used to fit the parameter using the known P, ET0 and observed runoff in the period of 1961-1970, then the fitted parameter was used to calculate the natural runoff in the following period. The explanation has been added in the P5L13-15 as "*Using the observed ET0, P and observed discharge, the parameter θ was calculated using the least-square data fitting method for the period 1961–1970, then the fitted parameter was used to reconstruct decadal natural runoff for the period 1971–2010.*"

"Some basins in the right most column of Figure 1" refers to arid basins and Hai River Basin. For those basins, Budyko framework usually performs poor. Here we only calculated the upstream natural runoff and assume there is no local-generated runoff in the downstream regions. The explanation of the runoff calculation in arid basins has been added in section 4.1 in P11L16-22 as "*What is more, previous studies proved that Budyko framework performed badly in arid and cold basins where snow and glacier melt contribute a lot to runoff. Here we found that Du et al. (2016) successfully applied a Budyko framework in Hei River Basin by dividing the basin into six sub-basins. They calibrated the model separately in different sub-basins and found the model performed quite well in the upper mountainous regions with little human interventions while the model was almost impossible to validate in downstream sub-basins. Thus we also divided the arid basins (including Hai, Shiyang, Hei and Tarim River Basins) into upper mountainous sub-basins and downstream sub-basins. The Fu-Budyko framework was directly applied in the mountainous*"

*sub-basins.*"

P6L24, it is very difficult to see which gauges are in the upstream and which gauges are in the downstream. The authors should think about how to present the locations of the gauges clearly.
Response: Thank you for the comments. The locations of upstream, middlestream and downstream and corresponding gauge stations have been marked in new Figure 1.

P6L25, can you explain why a gauge with a positive trend in rainfall can have a negative change (Fig. 3)?
Response: Here the rainfall's change was not calculated as the trend (mm/year) * year, instead it was calculated as the differences between two periods - 2000s and 1960s. Thus the change of a gauge was only dependent on the differences between 2000s and 1960s but not the trend. Meanwhile, Figure 3 has been deleted in the new manuscript.

P6L29, in northwest of China, such as Heihe, Tarim, river runoff is mostly contributed by snow melt. Is the framework suitable for these basin?
Response: The explanation of the runoff calculation in arid northwestern basins has been added in section 4.1 in P11L16-22 as "*What is more, previous studies proved that Budyko framework performed badly in arid and cold basins where snow and glacier melt contribute a lot to runoff. Here we found that Du et al. (2016) successfully applied a Budyko framework in Hei River Basin by dividing the basin into six sub-basins. They calibrated the model separately in different sub-basins and found the model performed quite well in the upper mountainous regions with little human interventions while the model was almost impossible to validate in downstream sub-basins. Thus we also divided the arid basins (including Hai, Shiyang, Hei and Tarim River Basins) into upper mountainous sub-basins and downstream sub-basins. The Fu-Budyko framework was directly applied in the mountainous sub-basins.*"

P6L25, P71, the authors gave subjectively the reasons for the trend (a significant increase in rainfall, recent global warming), I don't see any supporting analysis.
Response: All the trend analysis parts have been removed in the new manuscript according to the previous comment.

P7L11-15, from Figure 4, I can't see these interesting analyzes. And, please add the AI in this figure.
Response: The figure has been redrawn and the description of Figure 4 has been thoroughly rewritten in section 3.2.1.

P7L14-15, this is also too subjective.
Response: Figure 4 has been redrawn and the description of Figure 4 has been thoroughly rewritten in section 3.2.1.

P7L20, in Figure 5a, I suggest add an average of 1970s~2000s for each basin. Here, how did you define "continuously"? Obviously, WTA in the Yangtze, Pearl, Min River and Songhua did not increase monotonously.

Response: Figure 5 has been redrawn and the description of Figure 5 has been thoroughly rewritten in section 3.2.2.

P7L17-25, these results should be discussed to give the possible reasons.

Response: Section 4.2 fully discussed the changes of water policy in China, which shows that the excessive water use was encouraged in 1980s and 1990s while limited in 2000s. Please see the section 4.2 for details.

P8L5, Figure 7 is about water shortage, so I don't know why the authors were talking about surface water availability.

Response: Figure 7 has been replaced by Figure 4 and Figure 5, and the term "water shortage" was substituted by "per capita water availability". Please see section 3.2 for the details.

P8L11, water availability is determined by natural runoff, so I can't understand why population can affect water availability.

Response: We are sorry for the unclear description. Water availability is determined by natural runoff, while per capita water availability (FI) is related to population.

P8L19, from Figure 9a, I can't see the aggravation of water scarcity in China. This figure is not visual to show this aggravation trend.

Response: Figure 9 was replaced by Figure 7 and 8 and a new definition of water scarcity was used to identify the main driver of water scarcity changes. Please see the section 3.4 for the detailed explanation.

P8L25, water scarcity is defined with water stress and water shortage, here, why is it related to surface water availability?

Response: In the new manuscript, the term of "water shortage" was replaced by "per capita water availability", which is considered as better expression of "water shortage".

P8L28, fig. 9a and 9c cannot show this competition (at least I don't know how to interpret). And this paragraph was about water scarcity, but the authors were talking about water withdrawal. So it is very hard to understand these sentences.

Response: Figure 9 was replaced by Figure 7 and 8 and a new definition of water scarcity was used to identify the main driver of water scarcity changes. Please see the section 3.4 for the detailed explanation.

In Figure 8: in the Liao, Huai, and Qiantang, why were there no upstream, middle, and downstream?

Response: The explanation has been added in P7L4-9 as "*It is noted that only nine large basins were selected to analyze past changes in surface water scarcity in all three reaches (upper, middle and lower) because runoff data were not available in the downstream regions of Liao, Huai and Qiantang River Basins. For example, hydrological data at outlet station in Liao River Basin is available in 1984-2010; there were no hydrological data at outlet station in Huai River Basin; streamflow data were only available in tributary stations in Qiantang River Basin. For the above-mentioned three basins, we only used the available data from upper stream or tributaries for estimating WTA and FI.*"

P9L4-5, no analysis supporting the statement here.
Response: The climate-related content has been removed in the new manuscript.

P9L16, the possible impacts of the policies on water scarcity in all the basins were not fully discussed.
Response: The water policies from after the reform and opening were fully discussed in section 4.2 in the new manuscript, which shows that the excessive water use was encouraged in 1980s and 1990s while limited in 2000s. Please see the section 4.2 for details.

RESPONSE TO REVIEWER #2

General comments

Here, the authors presented a framework for quantifying the change in water scarcity at major river basins of China. Although, the study is interesting the methodology is not new and the manuscript is poorly written. For publishing purpose, the entire manuscript should be presented in a high quality format. The details of methodology is also not clear. In addition, the authors did not provided equal importance for all the objectives mentioned in the study.

Response: Thank you for reviewer #2's comments which greatly polished our manuscript. We used some new methods and thoroughly rewritten the manuscript. The detailed responses are as below.

Specific comments:

**Introduction**

The introduction should be improved with proper citations and sentences which shows the importance of the current study.

Response: The introduction has been rewritten and the review of literatures proved that our study is an important supplement for quantitative analysis of upstream-downstream water nexus.

Page 2 second paragraph is confusing. The sentences should be clear. Please add references for "A recent study has shown that the impact of anthropologic interventions on water scarcity is not always negative".

Response: The paragraph has been rewritten and the reference has been summarized in P2L16-20.

Line 21-22 (page2) is confusing. Please correct the sentence. Line 26 is not clear. Please rewrite the entire paragraph.

Response: The paragraph has been thoroughly rewritten. Please see the introduction part for the details.

Page 3. The presentation of objects is poor and not clear. Please write with specific reasoning. In addition, the sentence "The answers will provide experiences and lessons for global water resources management" is not matching here.

Response: The new objectives were proposed in revised manuscript in P3L24-29 as "In this study, we aim to answer following three questions, and provide experiences and lessons for global water resources management. They are:

i.   How surface water scarcity developed in upstream and downstream regions of selected basins in China during the past decades;

ii.  How to quantify the influence of upstream water use on downstream water scarcity; and

iii. What is the main driver for the change of China's water scarcity."

Overall, the introduction is too short and not clearly written. Please provide more

information on the importance of water scarcity analysis by using different indices such as, water stress and water shortage. Please try to link the importance of Fu-Budyko in water scarcity analysis in a river basin scale.

Response: We have thoroughly rewritten the introduction part which includes the points mentioned above. Please see the introduction part for the details.

**Materials and methods**

The manuscript needs more explanation on method section.

Response: More explanations has been added into the method section accordingly.

Starting the paragraph with 'because' is not recommended. In table 1 provide the lat/lon for gauge locations. Need more explanation on the section Hydrological data reliability.

Response: The first sentence has been corrected and lat/lon information has been added into Table 1. More information about the extracting and processing observed runoff has been provided to explain the data reliability. Please see the section 2.1.1 for details.

Line 29 - please replace e.g. by such as.

Response: All "e.g." has been replaced in the revised manuscript.

Page 4. The sentence "The steeper the catchment, the smaller was the parameter" is not clear. Need more explanation on the catchment parameter (theta) used in the study?

Response: In the new manuscript, the ordinary least squares method was used to fit the parameter. The explanation has been added in P5L13-15 as "*Using the observed ET0, P and observed discharge, the parameter θ was calculated using the least-square data fitting method for the period 1961–1970,thenthe fitted parameter was used to reconstruct decadal natural runoff for the period 1971–2010.*"

The equation 1 shows the Fu-Budyko framework, and it is a function of aridity index. But the authors did not mention it here. But in page 5 authors introduced the AI (aridity index). It will make confusion to the readers. Please rewrite the section accordingly.

Response: Thank you for pointing out the mistake. The introduction of AI has been moved to P5L5-9 as "*Where F(φ) is evaporation ratio, φ is the Aridicity index (AI), calculated from ratio of potential evapotranspiration (ET0) to precipitation (P) on annual scale, the θ parameter is related to catchment characteristics with the range of 1~∞. In this study, AI of each catchment was calculated at mean annual scale for the period of 1961-2010 and the catchments were classified into humid, semi-humid, semi-arid and arid for AI ranging from 0.375~0.75, 0.75~2, 2~5 and 5~12, respectively (Ponce et al., 2000; Arora, 2002).*"

Expand the unit mm/a (line 13)

Response: The unit of "mm/a" has been corrected to mm/year.

Hargreaves is not a suitable method for quantifying the potential ET hence it is only based on Tmax and Tmin. Please mention the drawback in the manuscript.

Response: The disadvantages of Hargreaves equation has been pointed and the combination of HG-ET0 and PM-ET0 was used to improve the accuracy. Please see P5L17-23 for details as "*Two equations – Hargreaves (HG) and Penman-Monteith (PM) – were used to estimate ET0 (Allen et al., 1998). The HG-ET0 was based on gridded dataset at monthly scale while PM-ET0 was based on pointed dataset at daily scale. The PM equation ranked as the best equation for estimating ET0 but the sparse distribution of climate stations limited its application in western China. The continuous spatial coverage of gridded dataset can provide full estimation of HG-ET0 in western China. However, large discrepancies between HG-ET0 and PM-ET0 were found in different regions over the world in previous studies (Temesgen et al., 2005; Gavilan et al., 2006; Trajkovic, 2007; Bautista et al., 2009; Sivaprakasam et al., 2011; Berti et al., 2014). Thus more accurate ET0 can be obtained by combining two estimations.*"

What does the value 17.8 indicates in the equation 2. Be more specific.

Response: The explanation of the parameters has been added in the revised manuscript in P5L27-28 as "*The standard values of empirical parameters are 0.0023, 17.8 and 0.5.*"

Then line 23-27 is not clear. Please rewrite.

Response: The paragraph has been rewritten in P6L6-10 as " *The monthly gridded HG-ET0 and daily pointed PM-ET0 were scaled up to annual value. At the annual scale, HG-ET0 was adjusted by multiplying the gridded coefficient (interpolated by the IDW method) as the ratio of the PM-ET0 to HG-ET0. The gridded annual precipitation was aggregated from the gridded monthly precipitation data and then adjusted by the point-scale data as mentioned above. The basin-scale annual P and ET0 were obtained by weighting average of grid data within each basin.*"

Page 5. Line 2-3. What is the basis of this classifications? Include references.

Response: The classification and its basis have been added in P5L6-9 as "*In this study, AI of each catchment was calculated at mean annual scale for the period of 1961-2010 and the catchments were classified into humid, semi-humid, semi-arid and arid for AI ranging from 0.375~0.75, 0.75~2, 2~5 and 5~12, respectively (Ponce et al., 2000; Arora, 2002).*"

The trend analysis section is not clear. Need more explanation including the equations used.

Response: In the new manuscript, the trend analysis relative contents have been removed.

Line 12-13 is not clear. Rewrite. Line 10-19 please rewrite. Please rewrite the section 'water stress and shortage'.

Response: The "Water stress and shortage" section has been rewritten as "Water stress and availability". Please see section 2.2.3 for details.

Line 17 Populations or population

Response: Populations has been changed to population.

The definition of WW is confusing. Please explain the $Q_{nat}$ and $Q_{obs}$ more specifically.

Response: The "WW" has been changed to "WU" and the detailed explanation was added in section 2.2.3. Please see the section for detailed explanation.

Page 6. Is it population count data or population data?

Response: It is population count data here.

Overall, the methodology section is not clearly written and confusing for the readers. Please improve the section.

Response: We thoroughly rewrote the method section and the explanation was clearer in the revised manuscript.

**Results**

The first sentence is not clear. What does the term sustainability indicates. Why did the authors calculate the correlation between observed and natural runoff? Need a clear explanation for this section.

Response: The improper term of "sustainability" has been replaced by "reliability" and the correlation coefficient was no longer calculated in the new manuscript. Please see the first paragraph of section 3.1 and P5L13-15 for the detailed explanation.

Page 6. The line 15-18 is not written well. Please improve the writing quality.

Response: The paragraph has been rewritten as "*The reliability of the Fu-Budyko framework in reconstructing annual natural discharge is summarized in Figure 2. The model captures well the fluctuations of observed discharge in both time and space during the simulation period of 1971-2010 in humid and semi-humid catchments, with small gaps between the observed and natural discharge (Fig. 2). Increasing gaps between the observed and natural discharge, however, are observed in semi-arid and arid basins, especially the Hai, Hei, Shiyang and Tarim River Basins. These gaps are regarded as water use from anthropologic activities.*"

Page 7. Line 18 shows that the authors selected only 9 large river basins for analysis. Please explain the reasons.

Response: The explanation for the selection of 9 river basins was added in P7L4-9 as "*It is noted that only nine large basins were selected to analyze past changes in surface water scarcity in all three reaches (upper, middle and lower) because runoff*

*data were not available in the downstream regions of Liao, Huai and Qiantang River Basins. For example, hydrological data at outlet station in Liao River Basin is available in 1984-2010; there were no hydrological data at outlet station in Huai River Basin; streamflow data were only available in tributary stations in Qiantang River Basin. For the above-mentioned three basins, we only used the available data from upper stream or tributaries for estimating WTA and FI.*"

The explanation for the questions "How did the imbalance in surface water scarcity develop between upstream and downstream regions? and What do we learn from China's water management strategies?" are not sufficient in the manuscript.
Response: The old objectives have been replaced by the new ones. Please see the last paragraph of Introduction section for the details. In the discussion section, we fully discussed the China's water policies since 1980s and linked the policies to the water scarcity. Please see the discussion section for the details.

Explain how the model framework is performing for different regions such as, snow regions in the manuscript.
Response: The suitability of Budyko framwork has been described in discussion section. Please see the section 4.1 for details.

The discussion on percentage decrease in surface water withdrawal is not clear. Please explain the possible reasons.
Response: The section 4.2 fully discussed the changes of China's water policies and their links to water scarcity. Please see the section 4.2 for the detailed explanation.

Page 9. Line 26-29 is not clear. The discussion section is not sufficient and well written.
Response: We have thoroughly rewritten the discussion section. The changes of water policies and their links to water scarcity has been fully discussed. Please see the discussion section for details.

[revised manuscript text omitted]

* * *
已下移 [1]: Note that Fu-Budyko ...
已下移 [2]: The Fu-Budyko framework...
已移动(插入) [3]

[revised manuscript text omitted]

**Yangtze**
| | Whole Basin | Upstream | Downstream |
|---|---|---|---|
| | 0 | 0 | 0 |
| | 0.73 | 0.52 | 0.87 |
| | 0.43 | 0.3 | 0.51 |

**Min**
| | Whole Basin | Upstream | Downstream |
|---|---|---|---|
| | 0 | 0 | 0 |
| | 0.03 | 0.63 | 0.88 |
| | 0.47 | 0.64 | 0.55 |

**Yellow**
| | Whole Basin | Upstream | Downstream |
|---|---|---|---|
| | 0 | 0 | 0 |
| | 0.58 | 0.55 | 0.53 |
| | 0.15 | 0.05 | 0.98 |

**Hei**
| | Whole Basin | Upstream | Downstream |
|---|---|---|---|
| | 0 | 0 | 0 |
| | 0.47 | 0.5 | 0.32 |
| | 0.14 | 0.48 | 0.55 |

**Tarim**
| | Whole Basin | Upstream | Downstream |
|---|---|---|---|
| | 0 | 0 | 0 |
| | 0.38 | 0.74 | 0.01 |
| | 0.93 | 0.09 | 0.62 |

**Xi (Pearl)**
| | Whole Basin | Upstream | Downstream |
|---|---|---|---|
| | 0 | 0 | 0 |
| | 0.48 | 0.71 | 0.61 |
| | 0.98 | 0.48 | 0.47 |

**Songhua**
| | Whole Basin | Upstream | Downstream |
|---|---|---|---|
| | 0 | 0 | 0 |
| | 0.01 | 0.02 | 0.02 |
| | 0.75 | 0.62 | 0.55 |

**Hai**
| | Whole Basin | Upstream | Downstream |
|---|---|---|---|
| | 0 | 0 | 0 |
| | 0.57 | 0.12 | 0.38 |
| | 0.46 | 0.57 | 0.51 |

**Shiyang**
| | Whole Basin | Upstream | Downstream |
|---|---|---|---|
| | 0 | 0 | 0 |
| | 0.97 | 0.74 | 0.6 |
| | 0.08 | 0.01 | 0.28 |

**WTA contribution in water scarcity change**
| | | |
|---|---|---|
| 0 | | |
| | Contribution$_{90\text{-}80}$ | |
| | | Contribution$_{00\text{-}90}$ |

---

## Referee Report (RR1)

Reconstructed natural runoff helps quantifying the relationship between upstream water use and downstream water scarcity in China's river basins:

Review: Thank you for incorporating the changes I had recommended during the first round of revision. I would like to thank and congratulate the authors for considerably improving the manuscript. I think the paper is now almost ready to be published in HESS after last minor corrections. In addition, it would be great if the authors can further improve the manuscript (writing).

**Line 1**: 'This is particular important for downstream areas where local-generated'… Please correct this sentence.

**Page 3, Line 5 – 9**: The paragraph is not clear. It would be good if the authors can clearly explain the aspects of water scarcity.

**Page 3**: It is difficult to compile historical data on long-term water use and the related water scarcity in China due to lack of data 10 accessibility or no long-term data available. The sentence is not clear. Please rewrite.

**Objective** iii. What are the main drivers contributing China's water scarcity… There is a missing preposition.

**Page 4:** 'population count data' – I think population data is the correct term. Change the heading to 'Population data'.

**Page 7**: 'zscore is calculated as'… The method is missing here? Please provide the method. In addition, the section 2.2.5 needs more explanation for a better understanding.

**Page 7**: "The model captures well the fluctuations of observed discharge in both time and space during the simulation period of 1971-2010 in humid and semi-humid catchments, with small gaps between the observed and natural discharge (Fig. 2). Increasing gaps between the observed and natural discharge, however, are observed in semi-arid and arid basins, especially the Hai, Hei, Shiyang and Tarim River Basins. These gaps are regarded as water use from anthropologic activities"

The above paragraph says the gap between modeled and observed flow is considered as water used for anthropogenic activities. But in the figure 2, some of the river basins experience a higher observed flow than modeled flow. Please give an explanation. How the authors can compare it with the real water usage from those (or all) watersheds? The models (statistical/hydrological) always try to overestimate the results. How the authors would incorporate this over estimation in the study?

**Figure 4**: Please provide the unit and explanation on the 'Y' axis. It would be easy to interpret.

**Page 8:** Please split the sentence "For most basins, 1980s is a critical period with rocketing WTA, for instance, 40% increase for Yangtze River Basin, 56% increase for Xi River Basin, 64% increase for Songhua River Basin, 52% increase for Yellow River Basin, 31% increase in Hai River Basin, 67% increase in Shiyang River Basin and 50% increase in national ranges".

**Page 9**: "Quantifying upstream-downstream water nexus". In the introduction section the authors did not provide any explanation on "water nexus". Please provide some explanation for this term.

**Page 10, Line 8**: 'The other basins have plenty available water but the excessive water use makes the Tarim River Basin, the downstream of Hei River Basin and the upstream of Songhua River Basin experiencing WTA stresses'. What are the probable sectorial water usage in that area? Pleas provide some explanations with references. In addition, please provide some real water usage examples for all the basins considered.

'Rocketed – is it possible to change the word with increased or some suitable word.

---

## Author Response (AR2)

Dear Professor Yuan,

We sincerely appreciate you and the two anonymous reviewers for all the kind help and valuable comments, which greatly polish our work. According to the comments, we clarified the definition of water scarcity and water stress, adjusted the ill-structured result section and revised the figures.

Especially, in the Acknowledge section, we added our thanks for three anonymous reviewers (the third helps a lot for the language) for their great help on the quality improvement of the work.

The point-to-point replies have been listed as follows and the responses were highlighted in blue for a distinction from the comments.

Best wishes,

Yonghui Yang

RESPONSE

Response to Reviewer #1
I think the revised manuscript was improved by the authors. However, the results part was still not well organized. I take section 3.2.1 for example. You firstly talked about WTA, the details for each basin were given (L12-13), but you started to descript FI in L14. Then you described WTA again, and last FI again! I strongly recommend you to separate the descriptions of WTA and FI, and don't mix up them! In figure 4, you should mark clearly (by vertical line?) which basins are humid, semi-humid…, so that the readers can follow up easily what you were saying in the figure. The use of the definitions were also confusing. You've already defined the water scarcity by WAT and FI, so please do not use per capita water availability (L14) again in the results, but use FI (In contrast, sometimes you used WTA instead water use to availability (L16)). Last, I strongly recommend you rename the abbreviations, i.e.., WTA to WS and FI to WA. Otherwise, the readers may be very easy to forget the meaning of the WAT (WAT seems water availability but actually refers to water stress in your definition, very misleading) and FI when they are reading the results.
Response: We appreciate the reviewer for the value comments and the confirmation of our efforts. According to the comments, we reconsidered the definitions of WTA and FI, as well as water scarcity and water stress. Then we redefined the "scarcity" as both WTA and FI were talking about, and the "stress" as only WTA or FI was involved in. Besides, the ill-structured result part and the figures mentioned by reviewer have been revised. Please see the following responses for the detailed revisions.

L20, you've defined WTA as water stress, so the "stress level" is misleading. The

readers may think you were only talking about WTA. Maybe change to scarcity level (because you were talking about both FI and WTA)? In summary, please use water stress, water availability, and water scarcity very strictly according to your definitions.

Response: Thank you very much for the insightful suggestion. Here the "stress" has changed to "scarcity" as suggestion. We redefined the "scarcity" as both WTA and FI were talking about, and the "stress" as only WTA or FI was involved in. The whole manuscript was reviewed and all the "stress" and "scarcity" have revised according to the rules.

L30, do not use per capita water availability but use FI or WTA suggested by me.

Response: All the "per capita water availability" has changed to "FI" in the context.

L26, how did you define the critical period? Why was 1980s critical when FI changed little? Apparently, you define the critical period by WTA, but why not FI?

Response: The "critical period" has changed to "turning point in WTA". The FI changed little because the water availability and population didn't change much. However, the reform and opening up policy made the water use increase dramatically. The paragraph is only related to WTA therefore we changed the "critical period" to "turning point".

P4L19, how was this gridded dataset generated? Please introduce the method briefly.

Response: The introduction of this gridded datasets has been added as "*The gridded datasets are produced by merging resampled GTOPO30 dataset and interpolated climate data from 2472 stations using TPS (Thin Plate Spline) method.*" in P4L21-22.

P5L27, how was Ra calculated?

Response: The explanation of Ra has been added in P5L29-30 as "*The calculation of extraterrestrial radiation $R_a$ is followed by FAO56 method (Allen et al., 1998).*"

P6L13, WAT refers to water stress, how about FI? According to P10L7-8, both of them were stress. So the authors should make very clear definitions on water stress, water availability, water scarcity, and do not mix up them in the results. According the title, I think WTA refers to water stress, and FI refers to water availability. Both of them constituted the water scarcity.

Response: Sorry for the confused statements of "water stress", "water availability" and "water scarcity". We think that WTA refers to water use and FI refers to water availability. Following the suggestion, we redefined the "scarcity" when both WTA and FI were talking about, and the "stress" when only WTA or FI was involved in as mentioned above.

P9, figure 5 was not cited by the main text.

Response: Figure 5 has been cited in the context in P9L3.

P7, section 3.1. This section is to evaluate the reliability of the framework. So the

comparison of simulated and observed annual runoff during 1971-2010 was meaningless (runoff was not natural!). Quantitative indicators were needed to evaluate the framework during 1961-1970.

Response: The reliability of Budyko framework has been quantified using biases between observed runoff and simulated runoff during calibration period of 1961-1970 in P7L25-27 and P8L1 as "*The model captures well the fluctuations of observed discharge in both time and space during the calibration period of 1961-1970 in all catchments, with biases (($\sum Qsim$-$\sum Qobs$)/ $\sum Qsim$) of 4.8%, 1.2%, 10%, -0.2%, -1.3%, 0.3%, 2.5%, -0.5%, 0.8%, 0.9%, -2.2% and 8.2% for Yangtze, Xi, Min, Qiantang, Huai, Songhua, Yellow, Liao, Hai, Hei, Shiyang and Tarim River Basins respectively (The calculation of biases for most stations using the downstream observed runoff while using upstream observed runoff for Hai, Hei, Shiyang and Tarim).*"

P9L2-3, I think stressed should be used for WTA because WTA refers to water stress. I understand that both WTA and FI can be described as stress according to your definition (P6L13-15), but it is very confusing when the readers try to link the stress to either WTA or FI. So please avoid describing FI as stressed (the water availability can be described as low or high?).

Response: We appreciate the helpful suggestion. Here we consider that stress can be indicated by both WTA and FI and described as water use stress or water availability stress. According to this, the definition of stress has been revised in section 2.2.3 as "*WTA indicates moderate or high water use stress when over 0.2 or 0.4, respectively (Vörösmarty et al., 2000). FI indicates moderate, high and extreme water availability stresses when it drops below 1,700, 1,000 and 500 m3 cap-1 year-1, respectively (Falkenmark, 1997).*"

P9L3, here, you were talking about FI in the upstream, why was it related to water use?

Response: Thank you for pointing out it. The FI-related content has been moved to the next paragraph. Now the first paragraph in P9 describes WTA and the second paragraph describes FI.

P10, figure 8 was not cited by the main text.

Response: Figure 8 has been cited in P10L20 now.

Section 3.2.2, this section was not well organized neither. You first introduce FI, and then WTA, and last FI again. Please move "from the FI…" to the P9L12.

Response: Thank you for the suggestion. The first paragraph has been rewritten to focus on WTA and the FI-related content has been moved to the next paragraph in P9L13-14.

P9L25, which basins?

Response: They are "Xi, Min, Yangtze and Songhua Basins" which have been added in P9L26.

Figure 6, how did you quantify the nexus? I guess by S1-S2? If I'm correct, please remove the S1 and S2, and only keep S1-S2.

Response: Yes, the difference in stress between two scenarios is used to quantify the influence of upstream water use on downstream water scarcity. Figure 6 has been redraw to keep only S1-S2 (S2-S1).

Figure 7, I think the most extreme case for water scarcity is that when WTA is maximum and FI is minimum. So please change the direction of the X axis (i.e., from 10500 to 0) to make it more clear.

Response: The x-axis in Figure 7 has been reversed according to the helpful suggestion.

Figure 8, I can't understand this figure. What do the 3*3 grids mean?

Response: Sorry for the confused figure. Now the figure 8 has change the form from grids figure to bar figure and the explanation for figure 8 in section 3.4 has been revised to make it clearer.

P11L6-9, it seems that the work by Du et al. (2016) was about human interventions but you were talking about snow and glacier melt. In the mountainous area, the Budyko framework performed poorly, so I don't understand you can apply the framework directly in the mountainous sub-basins (L12).

Response: Thank you for pointing out it. In fact we were also willing to talk about the arid basins with intensive human interference in northwestern China. Du et al. (2016) also mentioned the intensive human interference in arid region of northwestern China (pp. 394). Therefore we corrected the sentence into "*What is more, previous studies proved that Budyko framework performed badly in extremely arid environments where water systems are typically unclosed with intense human interference and irrigation, e.g., the northwestern China.*" in P11L9-10.

P13L29-32, please give the details of water scarcity (i.e., WTA and FI). Water scarcity here is not easy to understand.

Response: The sentences have been revised as "*Our results show that some river basins in China have experienced a dramatic increase of WTA stress from 1970s to 2000s due to the rapid increase of water use, which mainly occurs in northern basins. In 2000s, the increase of upstream WTA stress and the decrease of downstream WTA stress occurs simultaneously, which is probably caused by the increasing upstream water use and the consequent decrease of downstream water availability outpaced by the decrease of downstream surface water use.*" in P14L4-8.

Response to Reviewer #2

The current manuscript still has some grammatical errors, which needs to be improved by the authors. The Authors need to include more explanation on 'why the modeled flow is less than the observed flow for some river basins?'. In addition, the authors are expected to explain the sectorial water usage and its impacts in each river basins considered in the study.

Response: We really appreciate for your valuable comments which greatly polishes our work. According to the comments, we explain the possible reasons for the underestimation of natural runoff in this response letter. The statistical water use in different sectors from 2003-2017 and its change trend are simply described in section 3.4. And we further clarify the definition of WTA and FI. Please see the following for detailed responses.

Line 1: "This is particular important for downstream areas where local-generated..." Please correct this sentence.

Response: This sentence has been rewritten as "*This is particular important for downstream areas where upstream inflow is need to satisfy downstream water demand exceeding local-generated water resources.*" in Page 2, Line 1-3.

Page3, Line 5-9: The paragraph is not clear. It would be good if the authors can clearly explain the aspects of water scarcity.

Response: The paragraph has been revised and the explanation of water scarcity is more clear. "*Water scarcity can be divided into two aspects: water availability and water use. Water availability induced scarcity is a 'demographic-driven scarcity' when a large population compete for limited water resources, leading to disputes. Water use caused scarcity refers to a 'demand-driven scarcity' which potentially occurs with low population and high water use.*"

Page 3: "It is difficult to compile historical data on long-term water use and the related water scarcity due to lack of data accessibility or no long-term data available." The sentence is not clear. Please rewrite.

Response: The sentence has been written as "*It is difficult to get long-term water use and the related water scarcity in China due to data inaccessibility.*"

Objective iii. "What are the main drivers contributing China's water scarcity..." There is a missing preposition.

Response: The sentence has been revised as "*What is the main factor that has driven China's water scarcity change.*"

Page 4: "population count data" - I think population data is the correct term. Change the heading to "Population data".

Response: The "*population count data*" in heading of 2.1.3 and its content has changed to "*Population data*".

Page 7: "zscore is calculated as"...The method is missing here? Please provide the method. In addition, the section 2.2.5 needs more explanation for a better understanding.

Response: The method has been added and more explanation is provided as "*where ΔWTA and ΔFI indicate the absolute difference between two periods i and j in WTA and FI standardized (zscore) by subtracting the mean and dividing by the standard deviation. Every decade is compared with its previous decade to get the stress change, for example, 2000s - 1990s, 1990s - 1980s and 1980s is set to 0.*"

Page 7: "The model captures well the fluctuations of observed discharge in both time and space during the simulation period of 1971-2010 in humid and semi-humid catchments, with small gaps between the observed and natural discharge (Fig. 2). Increasing gaps between the observed and natural discharge, however, are observed in semi-arid and arid basins, especially the Hai, Hei, Shiyang and Tarim River Basins. These gaps are regarded as water use from anthropologic activities"

The above paragraph says the gap between modeled and observed flow is considered as water used for anthropogenic activities. But in the figure 2, some of the river basins experience a higher observed flow than modeled flow. Please give an explanation. How the authors can compare it with the real water usage from those (or all) watersheds? The models (statistical/hydrological) always try to overestimate the results. How the authors would incorporate this over estimation in the study?

Response: 1.The overestimation/underestimation in simulated runoff in specific year is considered as an intrinsic weakness of parameter calibration of hydrological models. The aim of parameter calibration is to minimize the biases between mean value of simulated and observed runoff. Therefore the fits of parameters are notoriously poor in producing extreme value by hydrological models. It reflects the need of further knowledge for extreme events in hydrological models.

2.The comparison between simulated water use and real water use is difficult here because 1) the simulated water use is only surface water consumption while real water use contains both surface and ground water consumption; 2) there is lack of long-term statistical water use data (from 2003-2017). However, the data is considered as reliable by domestic peer referee due to the reasonable FI value.

3. The higher value in natural runoff mainly occurs in the simulation period in semi-arid and arid basins. In fact, the model does not always overestimate runoff in calibration period. As mentioned in manuscript in P7L25-26, the biases between observed and simulated runoff $((\sum Qsim - \sum Qobs)/ \sum Qsim \times 100\%)$ is -0.2%, -1.3%, -0.5%, and -2.2% for Qiantang, Huai, Liao, and Shiyang River Basins respectively in period of 1961-1970.

Figure 4: Please provide the unit and explanation on the 'Y' axis. It would be easy to interpret.

Response: Thank you for the suggestion. The unit and explanation has been added on the y-axis for better understanding.

Page 8: Please split the sentence "For most basins, 1980s is a critical period with rocketing WTA, for instance, 40% increase for Yangtze River Basin, 56% increase for Xi River Basin, 64% increase for Songhua River Basin, 52% increase for Yellow River Basin, 31% increase in Hai River Basin, 67% increase in Shiyang River Basin and 50% increase in national ranges."

Response: The sentence has been revised as "*For most basins, 1980s is a critical period with rocketing WTA. The magnitude of increase is 40% for Yangtze River Basin, 56% for Xi River Basin, 64% for Songhua River Basin, 52% for Yellow River Basin, 31% for Hai River Basin, 67% for Shiyang River Basin and 50% for in national ranges.*"

Page 9: "Quantifying upstream-downstream water nexus". In the introduction section the authors did not provide any explanation on "water nexus". Please provide some explanation for this term.

Response: The "*water nexus*" has changed to "*the influence of upstream water use on downstream water scarcity*" in the context.

Page 10, Line8: "The other basins have plenty available water but the excessive water use makes the Tarim River Basin, the downstream of Hei River Basin and the upstream of Songhua River Basin experiencing WTA stresses". What are the probable sectorial water usage in that area? Please provide some explanations with references. In addition, please provide some real water usage examples for all the basins considered.

Response: The increase of water use in agricultural sector is the main factor to drive the WTA stress. This is explained in section 3.4 in Page 10 as "*
[revised manuscript text omitted]

